# RETHINKING FEDERATED AGGREGATION UNDER HETEROGENEITY: SCALABLE ENSEMBLES WITH OPEN-SET RECOGNITION

## ABSTRACT

Federated learning (FL) has gained widespread adoption as a privacy-preserving framework for distributed model training. However, it continues to face persistent challenges, most notably statistical heterogeneity and high communication cost. The current dominant paradigm in FL is consensus-driven averaging of model parameters across clients. Most recent methods, despite their innovations, remain anchored in repeated round averaging as the backbone of their design. The substantial communication overhead from repeated rounds is an obvious drawback, but another matter of debate is whether this approach can succeed under heterogeneous data, which forms the central focus of this paper. We argue that this prevailing approach fails to address heterogeneity. Using extreme label skew as a lens to expose its limitations, we demonstrate that even the most recent methods that ultimately rely on parameter averaging remain fundamentally limited in such settings. We instead advocate for an emerging alternative: ensemble-based FL with open-set recognition (OSR), which, by preserving client-specific models and selectively leveraging their strengths, directly mitigates the information loss and distortion caused by parameter averaging in heterogeneous settings. We consider this approach a principled path forward for addressing heterogeneity, substantiating our view through both theoretical analysis and extensive experiments. However, we acknowledge its primary limitation: the linear growth of ensemble size with client count, which hinders scalability. As a step forward in this direction, we introduce FedEOV, which incorporates improved negative sample generation to prevent shortcut cues, and FedEOV-pruned, which explores pruning as a solution to the scalability problem, rather than relying on distillation, thus avoiding the need for server-side data or additional training at the server. Our experiments across multiple datasets and heterogeneity settings confirm the superiority of our method, achieving an average improvement of 16.76% over the state-of-the-art ensemble baseline, FedOV, under extreme label skew and up to 102% over FedGF, the top-performing parameter averaging method. Furthermore, we show that pruned federated ensembles achieve performance on par with distilled ensembles, without any server-side data or training requirements, even when the latter is distilled using data from the same datasets. Code is available at: https://github.com/Anonymous6868-hue/FedEOV

## 1 INTRODUCTION

Real-world distributed machine learning scenarios often involve strict privacy constraints, where sharing raw data between parties is not permitted. Federated Learning (FL) has emerged as a popular paradigm in such settings, enabling clients to collaboratively train a global model without exchanging their private data Yang et al. (2019). A key objective in FL is to learn a model that generalizes well across all client distributions while keeping the confidentiality of individual data intact.

Standard FL framework, FedAvg, is based on parameter averaging where clients perform local training before sending it to a central server for averaging over multiple communication rounds to produce a global model McMahan et al. (2017). While simple and widely adopted, FedAvg relies on the assumption that client data is independent and identically distributed (IID), an assumption

that rarely holds in practice. In reality, federated systems often involve statistical heterogeneity, where clients have data drawn from different distributions. By averaging parameters, FedAvg seeks consensus across clients even when their local objectives diverge, making parameter averaging unreliable and slow in convergence, requiring massive communication rounds. Additional challenges arise from system heterogeneity, where clients differ in compute power and availability; model heterogeneity, where clients may use different architectures; and continual learning, where clients receive new data over time Pei et al. (2024); Criado et al. (2022).

Recently ensemble-based approaches have been proposed to address the communication efficiency and heterogeneity problems. While earlier works showed ensemble methods perform well under homogeneous data, more recent works have demonstrated their effectiveness to heterogeneous scenarios Diao et al. (2023). The state-of-the-art ensemble method, FedOV, uses open-set recognition (OSR) to identify an introduced unknown class while retaining the discriminative power of local models. This in a sense, naturally bypasses the issues like parameter misalignment Wang et al. (2020), and is inherently robust to statistical, system, and model heterogeneity. Notably, the performance of FedOV hinges on how effectively the OSR mechanism handles out of distribution shift at the local level. However, the primary limitation is that ensemble size grows linearly with the number of clients, making this approach impractical at scale.

Recent works such as FENS Allouah et al. (2024) and FedConcat Diao et al. (2024) propose hybrid approaches that combine elements of parameter averaging and ensemble. These methods correctly identify specialization, rather than consensus, as a key to handling client heterogeneity. However, both fall into the same core trap: they ultimately rely on parameter averaging to train the aggregation mechanism that combines specialized models. In doing so, they merely defer the heterogeneity problem to the final stage, where averaging, as a consensus mechanism, is inherently incapable of reconciling divergent client objectives. In this paper, we argue that parameter averaging should be avoided altogether. Instead, we theoretically show that open-set recognition during local training is sufficient for model aggregation, as it enables each client model to learn domain-specific information directly. The key idea is that when clients possess disjoint information, a specialization step within the solution is required to preserve each clients unique local knowledge. While this approach avoids global coordination and repeated communication entirely, it does lead to increased model size, a trade-off that we show can be managed through pruning. We term our method FedEOV: Federated Enhanced Open-set Voting. Our main contributions in this paper are:

- We provide a theoretical analysis of why parameter averaging is fundamentally limited in heterogeneous FL, particularly under extreme label skew, and why aggregation via OSR correctly preserves and integrates client-specific knowledge without the distortion caused by averaging. To our knowledge, prior works have not established this theoretical basis.

- We introduce FedEOV, an enhanced ensemble-based framework that improves the OSR mechanism through more principled negative sample generation. We demonstrate that through a small yet well-motivated change, FedEOV consistently outperforms FedOV(the strongest baseline under the extreme heterogeneity we study) in label-skewed scenarios.

- We identify model scalability as the key barrier to practical deployment of ensemble-based FL. To this end, we propose FedEOV-Pruned, a pruning-based strategy that compresses ensembles without requiring server-side data, unlike prior distillation-based methods. To the best of our knowledge, we are the first to propose pruning in ensemble-based FL, achieving significant model size reduction while maintaining competitive or superior accuracy even at high pruning levels, compared to distillation.

The remainder of the paper is organized as follows. In Section 2, we review related work in FL. Section 3 presents a comparative analysis of parameter averaging and ensemble-based methods. In Section 4, we introduce our proposed methods, FedEOV and FedEOV-Pruned. Section 5 covers our experimental setup/results, and we conclude in Section 6.

## 2 BACKGROUND AND RELATED WORK

**Non-IID Data in FL: Early Solutions and Theoretical Insights**: FL must confront data heterogeneity across clients, which significantly degrades its performance. The seminal FedAvg algorithm McMahan et al. (2017) performs well under IID data, but its accuracy degrades under non-IID

settings. When clients have divergent data distributions (e.g., different label proportions or label skew), the global model update from averaging local parameters can diverge from the true descent direction. Numerous works have documented this issue: for example, Zhao et al. (2018) showed that highly skewed label distributions can cause FedAvg's accuracy to drop by over 50%, and Li et al. (2020a) introduced FedProx to stabilize training via a proximal term. Even under IID data, averaging neural network weights can suffer from permutation inconsistency, leading to misaligned layers as noted by FedMA Wang et al. (2020). Mitigation strategies include correction of local updates (e.g., SCAFFOLD Karimireddy et al. (2020)), gradient harmonization Zhang et al. (2023), promoting flatter minima Qu et al. (2022a), explicit local–global alignment Li et al. (2021), and data sharing/augmentation. On the theoretical side, much early analysis of FedAvg focused on convex settings with guaranteed convergence under standard assumptions Li et al. (2020c), later extended to non-convex settings via bounded-dissimilarity assumptions in methods such as FedProx Li et al. (2020a) and FedDANE Li et al. (2020b). However, follow-up work Yuan & Li (2022) has shown that these assumptions conflict with the severe heterogeneity found in practice. Additional theoretical studies Diao et al. (2024); Allouah et al. (2024) have analyzed the fundamental limits of parameter averaging, including information-theoretic perspectives and quantification of the performance gap with alternative aggregation strategies. We refer readers to Appendix C.4 for a brief discussion on these error analyses.

**Recent Approaches to Label Skew in Federated Learning**: Despite these advances, there is still no single clear solution to the label skew problem, and a variety of techniques continue to be proposed. FedConcat Diao et al. (2024) clusters clients according to their label distributions, trains cluster-specific models via FedAvg, and constructs a global model by concatenating feature extractors across clusters while averaging only the classifier head. FedVLS Guo et al. (2025) addresses vacant-class scenarios by combining vacant-class distillation with logit suppression for non-local classes, thereby improving recognition of unseen labels while retaining parameter averaging. In addition, other approaches reflect different directions: FedLMD Lu et al. (2023) employs label-masking distillation to enhance minority-class learning, while FLea Xia et al. (2024) introduces obfuscated feature sharing with mixup-based augmentation under FedAvg. A particularly promising line of research focuses on sharpness-aware optimization, first explored in federated settings by FedSAM Qu et al. (2022a). Building on this idea, MoFedSAM Qu et al. (2022b) and the recent FedGF Lee et al. (2024) pursue flatter minima to alleviate client-drift and reduce the risk of model collapse under disjoint data.

**Ensemble-Based Approaches in Federated Learning**: Ensemble methods in FL were originally introduced to address the communication bottleneck, particularly in one-shot settings where each client trains locally and sends a model to the server only once Guha et al. (2019). Early designs simply averaged client models in a single round (one-shot FedAvg), but under severe heterogeneity this often yielded suboptimal results. This led to the alternative of combining *outputs* rather than weights, forming an ensemble at the server. While naive voting or averaging of predictions works for IID data, it fails in label-skewed settings, as models tend to misclassify unseen classes into seen ones, causing majority voting to collapse. Methods such as FEDBE Chen & Chao (2020), which treats global aggregation as a Bayesian ensemble over multiple global models, and FEDBOOST Hamer et al. (2020), which builds ensembles via weighted model averaging with theoretical guarantees for certain distributions, extended the ensemble concept but still faced this limitation. FEDOV Diao et al. (2023) addressed the problem by equipping each model with an open-set recognition mechanism that trains with synthetic outlier samples labeled as an *unknown* class, enabling models to abstain on unfamiliar inputs and improving ensemble decisions under heterogeneity. This OSR-based ensemble showed strong potential but has remained relatively underexplored. More recently, FEDCONCAT Diao et al. (2024) constructs a global model by concatenating the feature extractors of per-cluster models trained via FedAvg, averaging only the final classifier, thereby preserving specialized knowledge while still partially relying on parameter averaging, and FENS Allouah et al. (2024) learns a small neural aggregator at the server to fuse the outputs of client models in a stacked-generalization manner. Despite good performance under certain conditions and hyperparameters, these newer ensemble-style methods ultimately rely on parameter averaging to aggregate the unique information of the clients. This continued reliance reflects a common misunderstanding of the fundamental limitations of parameter averaging in FL, which motivates our theoretical analysis to clarify when and why averaging cannot be effectively used. For completeness, we review other One Shot FL categories in Appendix C.5, since they are not central to our analysis.

**Model Compression and Pruning in FL**: Although ensemble-based approaches were initially valued for reducing communication cost in federated learning and have recently shown strong potential in addressing heterogeneity, they carry a critical caveat: scalability. The scalability problem in FL has been recognized since early work such as Guha et al. Guha et al. (2019), where the cost of communicating and aggregating full models was shown to be a major bottleneck. Even if parameter averaging is avoided, a practical challenge for ensemble-based FL is the rapid growth in model size and deployment cost as the number of clients increases. Unlike FedAvg's single global model, an ensemble that retains all local models can become prohibitively large, with total parameters scaling linearly with the number of clients. This scalability issue makes vanilla ensembling impractical in large networks or on edge devices. A common strategy to address this has been knowledge distillation, explored since FedMD Li et al. (2019), which demonstrated that heterogeneous models can collaborate via public-data logit sharing without revealing architectures or private data. In the ensemble compression setting, the server uses a public or generated dataset to train a compact global model that imitates the ensemble's predictions. FedDF Lin et al. (2020) and related approaches exemplify this strategy, but they often assume access to auxiliary data at the server and risk sacrificing the diversity of the ensemble by collapsing it into a single model. Distillation essentially averages out the unique features of each client model, potentially losing the heterogeneity-based gains that ensembles offer. An alternative line of work explores *model pruning* to compress federated models. Li et al. Li et al. (2024) propose a client-side pruning approach where each client trains a local model and prunes less significant parameters before sending to the server, which then aggregates these slimmed models. Building on a similar intuition, our work (FedEOV-Pruned) applies pruning in the context of ensembles. As we will show, this strategy can compress an ensemble by an order of magnitude with minimal loss in accuracy, addressing the final barrier that prevents ensemble methods from becoming practical FL solutions at scale.

## 3 ANALYSIS OF AGGREGATION METHODS UNDER STATISTICAL HETEROGENEITY

In this section, we first build intuition for when and why parameter averaging and ensemble-based aggregation succeed or fail, starting from homogeneous scenarios and moving to increasingly skewed label distributions, with brief comments on communication cost. We then present theoretical bounds that formalize these observations. While our analysis centers on label skew, the advantages of ensembles with OSR extend to other FL challenges outlined in the introduction, including system heterogeneity, model heterogeneity, feature skew, and continual learning. Since these extensions are easier to see once the core case is understood, we present them separately in Appendix C.1 to keep the main discussion focused.

**In homogeneous setting, both parameter averaging and ensembling with OSR are effective, but differ in communication cost.** The first point to recall is that averaging, by its nature, accentuates commonalities and suppresses variability. This is why, under homogeneous data where each client approximates the same underlying function, parameter averaging is effective: the overlapping information enables convergence of the models toward a stable consensus. While the permutation invariance of neural networks may initially cause misalignment across client weights Wang et al. (2020), this typically resolves within a few rounds. Ensemble methods with OSR, in contrast, achieve strong performance in these settings using only a single communication round. In homogeneous settings, ensembling enhances generalization via a mixture-of-experts effect, leveraging model diversity across the clients. Also unlike parameter averaging, ensembles sidestep issues like permutation sensitivity and domain-specific misalignment, though at the cost of increased model size.

**Under mild label skew, parameter averaging can succeed given enough communication, with alignment-based methods offering more reliable performance.** In this setting, clients retain some label overlap, allowing global consensus to emerge over time. Even simple approaches like FedAvg may eventually converge, though often slowly and with reduced stability. Methods that explicitly aim to align client objectives or updates, such as SCAFFOLD, tend to perform better by correcting client drifts and accelerating convergence.

**Under extreme label skew, parameter averaging fails fundamentally, driven by two core issues: local drift and an information collapse caused by label partitioning.**

The first problem, local drift, is a well-known consequence of label skew, where clients converge to misaligned local optima. Although methods such as SCAFFOLD Karimireddy et al. (2020) recognize these challenges and attempt to realign client objectives to preserve the consensus-based formulation, these corrections are based on estimates of global gradients, which in turn rely on the very local gradients they aim to fix, creating a circular dependency. Debate continues around their utility under varying Dirichlet partitions, but in extreme label skew, where local gradients are entirely misaligned, these methods break down. Ensemble methods sidestep these elaborate alignment strategies by aggregating directly in function space, where such alignment is unnecessary as long as local models are trained to recognize out-of-distribution inputs.

The second failure is deeper: even with ideal optimization, heterogeneous label partitioning causes an information collapse. When clients see only a fraction of the global label space, the mutual information between model outputs and true global labels degrades with the number of labels per client, even under ideal training, as we will later show formally. This leads to trivial local optima; for example, when each client sees only a single label, a constant function minimizes the cross-entropy loss without learning anything meaningful for the global task. However, this collapse can be completely reversed by adding an abstention mechanism such as OSR. When clients are trained to abstain on unfamiliar inputs, the mutual information is fully recovered, as models must learn features that distinguish known from unknown. By preventing this collapse through OSR, the stage is set for functional aggregation.

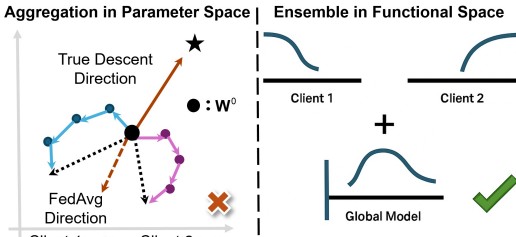

Figure 1: **Left:** Parameter-space averaging (e.g., FedAvg) can deviate significantly from the true descent direction, leading to unbounded error. **Right:** Functional-space aggregation (e.g., ensembles with OSR) preserves each client's specialization, enabling robust stitching of functions into a globally consistent model. Aggregation error here depends primarily on OSR performance.

These two failure modes, local drift and information collapse, motivate a formal analysis of the expected error in FL under extreme label skew, where we contrast parameter averaging with ensemble-based aggregation. Theorem 1 establishes bounds on mutual information, showing that label partitioning inevitably causes information loss even under ideal conditions, Theorem 2 establishes that ensemble aggregation with OSR constitutes an exact minimizer for an objective maximizing functional alignment of global model with all local models, and Theorem 3 provides error bounds that decompose into information, optimization, and local training errors for both parameter averaging and OSR-based ensembles. Complete proofs are provided in Appendix A.

**Theorem 1: Mutual Information under Idealized Training and Label Skew** *Consider a classification task over $N$ labels with uniform class priors. Each client is assigned a disjoint subset of $M$ labels, and models are trained under idealized conditions (perfect optimization, sufficient data). Let $Z_{nosr}$ denote the output of a model trained on disjoint labels without abstention, and $Z_{osr}$ denote the output of a model trained with OSR, where clients abstain on out-of-distribution inputs. Then, the mutual information between the model output and the true label satisfies:*

$$\textbf{Without OSR:} \quad I(Z_{\text{nosr}}; Y) \le \frac{M}{N} \log M, \tag{1}$$

$$\textbf{With OSR:} \quad I(Z_{\text{osr}}; Y) \le \log N. \tag{2}$$

**Discussion:** The quantity $I(Z_{\text{nosr}}; Y)$ is strictly bounded by the fraction of the label space each client observes, increasing monotonically with $M$, and reaching its maximum $\log N$ only in the centralized case $M = N$. In contrast, training with OSR fully recovers the information about the global label space, achieving the optimal bound $\log N$ regardless of how labels are partitioned across clients.

**Theorem 2: Optimal Functional Aggregation** *Let $\{f_c\}_{c=1}^{C}$ be local client models trained with OSR, where each model outputs a class probability vector over its known labels plus an abstention token $\perp$. Define the confidence weight for client $c$ on input $x$ as $\alpha_c(x) = 1 - f_c(x)_\perp$. Then, the*

*global model $f^*$ that minimizes the following confidence-weighted functional alignment objective:*

$$\mathcal{L}(f^*) = \sum_{c=1}^{C} \mathbb{E}_{x \sim \mathcal{D}_c} \left[ \alpha_c(x) \cdot \|f^*(x) - f_c(x)\|^2 \right] \tag{3}$$

*has the following solution, which acts as a confidence-weighted ensemble of the local models:*

$$f^*(x) = \frac{1}{\sum_{c=1}^{C} \alpha_c(x)} \sum_{c=1}^{C} \alpha_c(x) \cdot f_c(x) \tag{4}$$

**Discussion:** The global objective $\mathcal{L}(f^*)$ is convex in $f^*$, and the solution above is the exact global minimizer in closed form. This is why it requires only a single communication round and guarantees optimal alignment in the output space, with residual error determined solely by the accuracy and abstention behavior of the local models. In contrast, FedAvg operates in parameter space and performs only an approximate gradient descent step, which explains its iterative nature. However, this approximation breaks down in highly non-IID scenarios, where local objectives diverge significantly. As a result, the aggregation step is no longer a true descent direction, and the associated error becomes unbounded. Ensemble methods with OSR avoid this failure by aggregating directly in function space, effectively stitching together the specialized knowledge of local models using information about where each model is valid.

**Theorem 3: Expected Test Error under Extreme Label Skew** *Let $\mathcal{E}_{avg}$ and $\mathcal{E}_{ens}$ denote the expected test error of a global model obtained via parameter averaging and ensemble aggregation with OSR, respectively, in a federated setting with disjoint label partitions. Let $w_c(x) = \frac{\alpha_c(x)}{\sum_{c=1}^{C} \alpha_c(x)}$ denote the normalized confidence weights used in ensemble aggregation. Then:*

$$\textbf{Parameter Averaging:} \quad \mathcal{E}_{\text{avg}} \leq \underbrace{\sum_{c=1}^{C} \mathbb{E}[\ell(f_c(x), y)]}_{\text{Local training error}} + \underbrace{\varepsilon_{\text{align}}}_{\text{Alignment error}} + \underbrace{\left(\log N - \tfrac{M}{N} \log M\right)}_{\text{Label distribution error}} \tag{5}$$

$$\textbf{Ensemble with OSR:} \quad \mathcal{E}_{\text{ens}} \leq \underbrace{\sum_{c=1}^{C} \mathbb{E}_{(x,y) \sim \mathcal{D}_c} \left[ w_c(x) \cdot \ell(f_c(x), y) \right]}_{\text{Local + OSR error}} + \underbrace{0}_{\text{Label dist \& alignment error}} \tag{6}$$

**Implication:** In ensemble aggregation, the only source of error arises from local model training and the performance of the OSR, which controls confidence weighting $w_c(x)$. In contrast, parameter averaging introduces additional error through the aggregation of model parameters, which is not a true descent direction, especially when local objectives differ. Most prior works focus primarily on this error caused by misalignment, with a range of analyses attempting to bound it under various assumptions. We discuss these efforts and the alignment error term in greater detail in Appendix C.4. However, the conditions under which these theoretical bounds hold are rarely satisfied in practice. As a result, the misalignment error remains substantial in realistic federated settings. Moreover, parameter averaging incurs an additional large error due to disjoint label distributions, leading to much higher test error in practice. Therefore, under extreme label skew, we consistently observe $\mathcal{E}$avg $> \mathcal{E}$ens.

## 4 FEDEOV: FEDERATED ENHANCED OPEN-SET VOTING

We propose FedEOV, a one-shot ensemble method for FL that enhances OSR and addresses scalability via model pruning. We introduce a more structured and effective negative sample generation process that improves robustness to unseen classes. To mitigate the ensemble size growth inherent to one-shot ensembling, we further introduce FedEOV-Pruned, a client-side, data-driven pruning scheme that significantly compresses each local model with minimal accuracy degradation.

**Enhancing Open-Set Recognition with Progressive Augmentations:** In OSR, the goal is to ensure that models abstain confidently on inputs from unseen classes. FedOV approaches this by introducing synthetic negatives using cut-paste operations, region erasure, and adversarial perturbations using the

Fast Gradient Sign Method (FGSM) Goodfellow et al. (2014). However, these augmentations often leave behind structured low-level artifacts (e.g., hard edges or textures) that models can overfit to; this, in turn, can enable trivial rejection of synthetic samples without learning semantically meaningful boundaries. We refine this mechanism by employing a progressive three-stage training strategy, designed to remove such shortcut cues. The first two stages mirror FedOV augmentations: standard region erasure and cut-paste operations to introduce coarse disruptions, followed by untargeted FGSM adversarial samples to confuse decision boundaries. In the final stage, we introduce harder samples in the form of shuffled-patch augmentations with smoothed transitions. These transitions eliminate shortcut cues such as sharp edges, preserving textural consistency while disrupting global semantics, forcing the model to learn more meaningful representations. Appendix D provides the full algorithm, and the implementation is available on GitHub through the link in the abstract.

**Ensemble Scalability via Pruning:** A core challenge in one-shot ensemble FL is that the global model size grows linearly with the number of clients. Knowledge distillation has been proposed as a remedy, but it typically requires server-side data and often dilutes model diversity, making it ill-suited for many FL settings. Instead of relying on distillation, we note that a dense model trained in a centralized manner can achieve strong performance with fixed capacity. This motivates the hypothesis that a carefully pruned ensemble (effectively a convex combination of multiple models) should, with the same overall parameter budget, retain sufficient capacity to perform well.

To test this, we employ an iterative lottery ticket pruning scheme executed locally on each client. At fixed intervals (e.g., every 10 epochs), each model undergoes per-layer pruning based on activation magnitudes. Surviving weights are reset to their original pre-training values before training resumes. Repeating this process gradually reduces model size while preserving essential representational capacity. Our aim is to demonstrate that pruning, unlike distillation, which often conflicts with FL constraints, not only remains feasible but also tends to sustain higher accuracy. Further details of the pruning procedure are provided in Appendix D, with a discussion of training and inference costs, along with the limitations of distillation, in comparison to distillation in Appendix C.2.

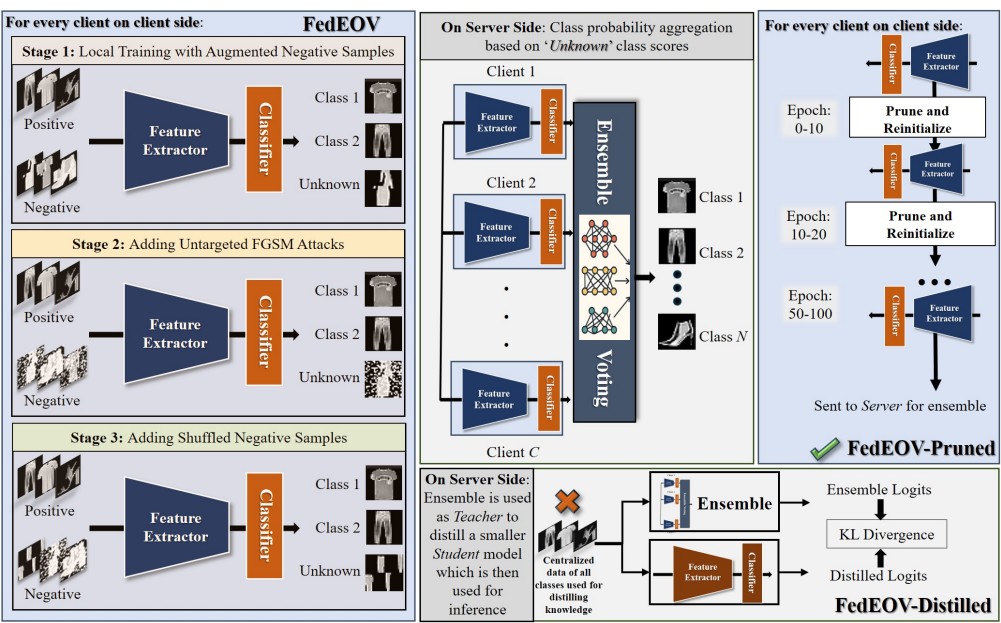

Figure 2: Overview of FedEOV and its scalable extensions. **Left:** *FedEOV client-side training* is carried out in three stages involving negative augmentation, adversarial attacks, and shuffled negatives to enable OSR. **Middle:** *Server-side ensemble voting* aggregates client predictions based on unknown class confidences to infer the true label. **Right:** *FedEOV-Pruned* applies layer-wise pruning and reinitialization on clients to reduce model size before sending it to server for ensemble. **Bottom:** *FedEOV-Distilled* compresses the ensemble into a student model trained with distillation using centralized class-balanced data.

## 5 EXPERIMENTS

### 5.1 MAIN RESULTS

We primarily evaluate FedEOV under the *extreme label skew* setting, where each client has disjoint class labels, representing the most challenging non-IID scenario (see Table 2). For completeness, we also report results on two additional settings: a standard *label skew* scenario using Dirichlet sampling ($\alpha = 0.1$) (Table 3) and a *homogeneous* IID setting (Table 4). The parameter counts across methods and client numbers are summarized in Table 1.

Table 1: Parameter Count Comparison Across Methods

| Clients | FedAvg | SCAFFOLD | FedConcat | MoFedSAM | FedGF | FedOV | FedEOV* | FedEOV-Distilled | FedEOV-Pruned* |
|---|---|---|---|---|---|---|---|---|---|
| 5 | 150K | 150K | 750K | 150K | 150K | 750K | 750K | 150K | 150K |
| 10 | 150K | 150K | 750K | 150K | 150K | 1.5M | 1.5M | 150K | 150K |
| 20 | 150K | 150K | 750K | 150K | 150K | 3M | 3M | 150K | 150K |

Table 2: Performance Comparison of Federated Learning Methods (Extreme Heterogeneity)

| # | Dataset | FedAvg | SCAFFOLD | FedConcat | MoFedSAM | FedGF | FedOV | FedEOV* | FedEOV-Distilled | FedEOV-Pruned* |
|---|---|---|---|---|---|---|---|---|---|---|
| 5 | MNIST | 81.56 | 82.97 | 83.88 | 93.69 | **93.95** | 83.77 | 87.62 | 67.2 | 85.69 |
| | FMNIST | 66.05 | 64.68 | 63.34 | 75.31 | **75.57** | 68.89 | 74.0 | 61.8 | 75.41 |
| | SVHN | 60.58 | 63.91 | 51.36 | 44.0 | 53.37 | 51.5 | **77.74** | 73.96 | 75.62 |
| | CIFAR-10 | 49.03 | 49.05 | 46.49 | 47.68 | 48.22 | 69.09 | **80.83** | 67.53 | 68.68 |
| | CIFAR-100 | 29.54 | 29.5 | 3.81 | 28.62 | 29.98 | 86.14 | **87.69** | 62.38 | 67.16 |
| | Tiny-ImageNet | 16.97 | 14.94 | 0.45 | 15.03 | 15.58 | 65.76 | **73.06** | 27.55 | 40.86 |
| 10 | MNIST | 45.29 | 50.44 | 40.66 | 60.76 | 61.1 | 68.42 | **85.61** | 45.72 | 73.79 |
| | FMNIST | 60.3 | 60.31 | 28.84 | 64.62 | 64.73 | 64.64 | **73.12** | 54.69 | 65.22 |
| | SVHN | 16.75 | 12.49 | 9.13 | 19.07 | 19.07 | 37.29 | **77.86** | 67.61 | 74.33 |
| | CIFAR-10 | 22.45 | 21.71 | 18.75 | 25.66 | 25.96 | 43.27 | **64.06** | 48.27 | 49.59 |
| | CIFAR-100 | 20.5 | 20.51 | 3.44 | 19.03 | 20.07 | 77.07 | **81.89** | 55.96 | 54.58 |
| | Tiny-ImageNet | 11.9 | 11.46 | 1.04 | 11.01 | 11.13 | 73.09 | **82.01** | 26.14 | 32.87 |
| 20 | MNIST | 25.04 | 43.32 | 38.73 | 60.87 | 61.25 | 85.06 | **89.69** | 74.59 | 89.63 |
| | FMNIST | 38.56 | 61.2 | 36.12 | 65.23 | 65.19 | 71.5 | 76.42 | 70.71 | **79.19** |
| | SVHN | 19.05 | 12.57 | 9.47 | 14.41 | 14.41 | 63.24 | **76.47** | 71.57 | 73.46 |
| | CIFAR-10 | 23.63 | 24.22 | 18.83 | 25.95 | 26.09 | 62.77 | **71.9** | 63.83 | 58.93 |
| | CIFAR-100 | 12.87 | 12.7 | 3.69 | 12.11 | 12.51 | 85.4 | **93.56** | 58.59 | 55.5 |
| | Tiny-ImageNet | 10.45 | 10.21 | 1.53 | 7.02 | 7.53 | 64.56 | **72.64** | 25.31 | 26.13 |

Table 3: Performance Comparison of Federated Learning Methods (Non-IID (Dirichlet 0.1))

| # | Dataset | FedAvg | FedOV | FedEOV* | FedEOV-Distilled | FedEOV-Pruned* |
|---|---|---|---|---|---|---|
| 5 | MNIST | **93.69** | 93.07 | 90.15 | 90.84 | 87.56 |
| | FMNIST | 76.16 | **81.82** | 80.98 | 79.57 | 70.23 |
| | SVHN | 71.88 | 72.67 | **79.91** | 77.31 | 79.16 |
| | CIFAR-10 | 54.36 | 80.91 | **83.53** | 76.47 | 69.91 |
| | CIFAR-100 | 34.93 | 88.85 | **90.04** | 66.79 | 73.78 |
| | Tiny-IN | 26.38 | **71.42** | 69.74 | 33.8 | 44.09 |
| 10 | MNIST | 85.48 | **89.5** | 87.93 | 87.41 | 87.54 |
| | FMNIST | 71.85 | 79.76 | 80.76 | 78.41 | **85.3** |
| | SVHN | 50.78 | 78.49 | **81.45** | 80.01 | 67.26 |
| | CIFAR-10 | 43.27 | 73.28 | **82.06** | 74.19 | 63.35 |
| | CIFAR-100 | 25.81 | 90.11 | **92.77** | 70.39 | 61.68 |
| | Tiny-ImageNet | 18.54 | 77.94 | **79.13** | 31.97 | 40.9 |
| 20 | MNIST | 59.55 | 92.67 | **93.06** | 92.25 | 92.5 |
| | FMNIST | 62.96 | 81.09 | 81.98 | **82.42** | 80.04 |
| | SVHN | 17.53 | 76.95 | **81.81** | 80.35 | 74.89 |
| | CIFAR-10 | 40.59 | 77.4 | **81.83** | 75.26 | 66.96 |
| | CIFAR-100 | 19.2 | 92.93 | **95.3** | 74.18 | 63.07 |
| | Tiny-IN | 13.62 | 82.66 | **87.43** | 33.98 | 34.76 |

Table 4: Performance Comparison of Federated Learning Methods (Homogeneous)

| # | Dataset | FedAvg | FedOV | FedEOV* | FedEOV-Distilled | FedEOV-Pruned* |
|---|---|---|---|---|---|---|
| 5 | MNIST | 95.31 | **99.24** | 99.08 | 98.96 | 98.97 |
| | FMNIST | 81.36 | **93.03** | 92.41 | 91.54 | 90.98 |
| | SVHN | 50.48 | **92.62** | 90.99 | 90.09 | 89.32 |
| | CIFAR-10 | 79.19 | 91.2 | **92.33** | 88.78 | 84.3 |
| | CIFAR-100 | 48.57 | **90.16** | 89.88 | 88.94 | 67.7 |
| | Tiny-ImageNet | 34.07 | **71.35** | 69.25 | 33.78 | 41.69 |
| 10 | MNIST | 91.77 | 98.94 | 98.87 | 98.7 | **99.14** |
| | FMNIST | 76.3 | 91.8 | 91.35 | 90.62 | **93.35** |
| | SVHN | 19.15 | 90.9 | 89.58 | 88.78 | **91.79** |
| | CIFAR-10 | 69.06 | 71.44 | **89.8** | 86.58 | 77.16 |
| | CIFAR-100 | 32.88 | 90.34 | **92.09** | 71.04 | 61.68 |
| | Tiny-ImageNet | 22.32 | 79.64 | **79.83** | 33.61 | 37.75 |
| 20 | MNIST | 88.84 | 98.37 | **98.42** | 98.3 | 97.49 |
| | FMNIST | 74.32 | **90.14** | 89.71 | 89.22 | 86.47 |
| | SVHN | 19.07 | 87.96 | **88.18** | 87.32 | 83.41 |
| | CIFAR-10 | 59.04 | 92.2 | **95.51** | 92.06 | 76.12 |
| | CIFAR-100 | 22.1 | 97.25 | **98.23** | 84.69 | 75.01 |
| | Tiny-ImageNet | 12.25 | 84.62 | **87.2** | 36.31 | 30.53 |

**Baseline Methods:** We evaluate a range of federated learning methods spanning different aggregation paradigms. For standard parameter averaging approaches, we include FedAvg, SCAFFOLD, MoFedSAM, and FedGF, using the hyperparameters provided in the original implementations of MoFedSAM and FedGF. For ensemble-based methods, we compare our FedEOV with FedOV which is state-of-the-art in ensemble methods. We also evaluate FedConcat, a hybrid ensemble approach that incorporates both ensemble aggregation and parameter averaging, using the default clustering hyperparameter from its original implementation. Additionally, we assess the efficacy of different compression methods at the same parameter budget, by evaluating compressed variants of FedEOV: FedEOV-Pruned and FedEOV-Distilled. To demonstrate the maximum potential of distillation, we perform server-side distillation using IID data sampled from the actual dataset. We adjust the pruning ratio per client setting to match the budget.

**Federated Configuration and Datasets:** Experiments span a range of standard vision benchmarks. For parameter averaging and hybrid methods, we train for 100 communication rounds across all datasets. On smaller datasets (MNIST, Fashion-MNIST, SVHN), we use 5 local epochs per round, while for larger datasets (CIFAR-10, CIFAR-100, Tiny-ImageNet), we use 10 local epochs per round. For ensemble-based methods, which require no communication rounds, we train for 10 local epochs

on smaller datasets and 100 local epochs on larger datasets. All experiments are conducted across 5, 10, and 20 client configurations.

**Default Setup:** All models use a simple Convolutional Neural Network (CNN) with two convolutional layers and one fully connected layer, trained with a learning rate of 0.001. Each experiment is repeated across multiple random seeds to ensure statistical reliability, with most experiments conducted over 5 seeds and mean performance reported. All experiments are run on a single NVIDIA RTX 4090 GPU.

**Additional Experiments.** Appendix B presents additional results, including experiments across additional heterogeneity scenarios (feature skew), larger CNN architectures, varying Dirichlet parameters, high-client-count configurations for datasets with numerous classes, and comprehensive comparisons with a broader range of FL methods.

## 5.2 RESULT ANALYSIS

**Ensemble-based methods consistently outperform parameter averaging and hybrid methods.** Performance across all methods remains reasonable in homogeneous and non-IID (Dirichlet 0.1) settings, but ensemble methods demonstrate superior accuracy while requiring significantly less communication overhead, as explained by the theoretical foundations discussed in Section 3. However, extreme label skew serves as the ultimate stress test: ensemble-based approaches demonstrate resilience while all other methods experience severe performance degradation. This gap becomes particularly pronounced on more challenging datasets such as CIFAR-10, CIFAR-100, and Tiny-ImageNet, where the inherent challenges of extreme heterogeneous data expose the core weaknesses of parameter averaging approaches. Notably, FedConcat appears especially compromised in these extreme settings, which we attribute to its sensitivity to clustering parameters that fail to generalize across our particular testing conditions.

**FedEOV consistently outperforms FedOV in label skew settings.** This overall average gain is 16.76% and can be attributed to our enhanced OSR strategy. In contrast, under homogeneous data distributions, FedEOV offers no advantage over FedOV, which means that effective OSR is not critical when all clients have access to all classes and can make confident predictions across the label space.

**FedEOV-Pruned achieves comparable performance to distilled models despite operating under more realistic assumptions.** In fact, under extreme heterogeneity pruning shows a 12.04% average gain over distillation. This result is particularly significant given that the distilled variant requires access to centralized server-side data (an unrealistic assumption in many FL deployments). In contrast, pruned models achieve competitive accuracy without requiring such privileged information or additional com-

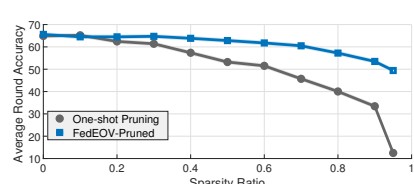

Figure 3: Accuracy vs. pruning ratio.

putational overhead. Notably, on less complex datasets, pruning can actually improve performance beyond the original model. While very high pruning can degrade accuracy, its key advantage lies in providing a dial to balance performance and compression by adjusting the pruning ratio, as illustrated in Figure 3; iterative pruning is particularly effective under extreme label skew (CIFAR-10). These results suggest that in realistic federated settings, where central data is unavailable, heavily pruned ensemble models offer a compelling alternative to distillation. Training and inference cost comparisons are discussed in Appendix C.2.

## 6 CONCLUSION

In this paper, we considered the problem of statistical heterogeneity in FL framework and analyzed the emerging paradigm of ensemble-based FL with OSR in comparison to dominant consensus-driven parameter averaging across client models. We have shown that ensemble with OSR mitigates information loss caused by data heterogeneity where many state-of-the-art methods struggle. Building on our analysis, we introduced FedEOV, which improves performance of ensemble-based FL by enhancing the OSR mechanism, and FedEOV-Pruned, which demonstrates that pruning is a viable solution to the scalability challenge inherent to ensemble methods.

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
