## APPENDIX A: PROOFS

### A.1 THEOREM 1: MUTUAL INFORMATION WITHOUT OSR

**Assumptions:** We define the following assumptions for the analysis of mutual information in federated learning under extreme label skew:

- **A1: Uniform Label Prior:**
  The global label space is $\mathcal{Y} = \{1, 2, \ldots, N\}$, and the true label $Y$ is drawn uniformly:

$$P(Y = y) = \frac{1}{N}, \quad \forall y \in \mathcal{Y}.$$

- **A2: Disjoint Label Partitions:**
  Each client $k$ is assigned a disjoint subset of labels $\mathcal{Y}_k \subset \mathcal{Y}$, such that:

$$|\mathcal{Y}_k| = M, \quad \mathcal{Y}_i \cap \mathcal{Y}_j = \emptyset \text{ for } i \neq j, \quad \bigcup_k \mathcal{Y}_k = \mathcal{Y}.$$

- **A3: Deterministic Prediction on In-Distribution Labels:**
  For any label $y \in \mathcal{Y}_k$, the client's model $f_k$ outputs the correct label with probability 1:

$$P(Z = y \mid Y = y) = 1.$$

- **A4: Uniform Prediction on Out-of-Distribution Labels (No OSR):**
  For $Y \notin \mathcal{Y}_k$, the model predicts a label from $\mathcal{Y}_k$ uniformly at random:

$$P(Z = z \mid Y \notin \mathcal{Y}_k) = \frac{1}{M}, \quad \forall z \in \mathcal{Y}_k.$$

- **A5: No Feature Shift:**
  Inputs $x$ are drawn i.i.d. and independently of the client. The analysis focuses solely on label-distribution skew.

Let assumptions (A1)–(A5) hold, then the mutual information between the model output $Z$ and the true label $Y$, for a client trained without OSR, is:

$$I(Z; Y) = \frac{M}{N} \log M.$$

PROOF

By the definition of mutual information,

$$I(Z; Y) = H(Y) - H(Y \mid Z).$$

From assumption (A1), the label prior is uniform over $N$ classes, so:

$$H(Y) = \log N.$$

We compute the conditional entropy $H(Y \mid Z)$ using the law of total probability:

$$H(Y \mid Z) = \sum_{z \in \mathcal{Y}_k} P(Z = z) \cdot H(Y \mid Z = z).$$

From assumptions (A3) and (A4), the joint distribution $P(Y, Z)$ is:

- For $y \in \mathcal{Y}_k$, $Z = y$ deterministically: $P(Y = y, Z = y) = \frac{1}{N}$.

- For $y \notin \mathcal{Y}_k$, the model chooses $Z \in \mathcal{Y}_k$ uniformly: $P(Y = y, Z = z) = \frac{1}{N} \cdot \frac{1}{M} = \frac{1}{NM}$, for all $z \in \mathcal{Y}_k$.

Thus, for each $z \in \mathcal{Y}_k$, the marginal distribution is:

$$P(Z = z) = \frac{1}{N} + \frac{N - M}{N} \cdot \frac{1}{M} = \frac{1}{M},$$

which implies $Z$ is uniformly distributed over $\mathcal{Y}_k$.

The posterior distribution for each fixed $z \in \mathcal{Y}_k$ is:

$$P(Y = z \mid Z = z) = \frac{\frac{1}{N}}{\frac{1}{M}} = \frac{M}{N}, \quad P(Y = y \mid Z = z) = \frac{\frac{1}{NM}}{\frac{1}{M}} = \frac{1}{N}, \quad \forall y \notin \mathcal{Y}_k.$$

Therefore, the conditional entropy at each $Z = z \in \mathcal{Y}_k$ is:

$$H(Y \mid Z = z) = -\frac{M}{N} \log\left(\frac{M}{N}\right) - (N - M) \cdot \frac{1}{N} \log\left(\frac{1}{N}\right).$$

Simplifying:

$$\begin{aligned}
H(Y \mid Z = z) &= -\frac{M}{N}(\log M - \log N) + \left(1 - \frac{M}{N}\right) \log N \\
&= -\frac{M}{N} \log M + \frac{M}{N} \log N + \left(1 - \frac{M}{N}\right) \log N \\
&= \log N - \frac{M}{N} \log M.
\end{aligned}$$

Since this is the same for every $z \in \mathcal{Y}_k$ and $Z$ is uniform over this set, we have:

$$H(Y \mid Z) = \log N - \frac{M}{N} \log M.$$

Thus,

$$I(Z; Y) = H(Y) - H(Y \mid Z) = \log N - \left(\log N - \frac{M}{N} \log M\right) = \frac{M}{N} \log M.$$

$\blacksquare$

THEOREM 1: MUTUAL INFORMATION WITH OSR

Let assumptions (A1), (A2), (A3), (A5) hold, and replace (A4) with (B4) as defined below. Then the mutual information between the ensemble model output $Z_{\text{ens}} = (Z_1, \ldots, Z_C)$ and the true label $Y$, under abstention-based ensemble aggregation, is:

$$I(Z_{\text{ens}}; Y) = \log N.$$

ADDITIONAL ASSUMPTION FOR OSR SETTING

- **(B4) Deterministic Abstention on OOD Labels (OSR):**
  For $Y \notin \mathcal{Y}_k$, client $k$ outputs a special abstention token $\perp$, i.e.,

  $$P(Z_k = \perp \mid Y \notin \mathcal{Y}_k) = 1, \quad P(Z_k = z \mid Y \notin \mathcal{Y}_k) = 0, \quad \forall z \in \mathcal{Y}_k.$$

PROOF

Each client $k$ outputs either a class label $z \in \mathcal{Y}_k$, if $Y \in \mathcal{Y}_k$, or abstains with $Z_k = \perp$, if $Y \notin \mathcal{Y}_k$. By assumption (A2), the global label space $\mathcal{Y}$ is partitioned disjointly across clients, so for every $y \in \mathcal{Y}$, there exists a unique client $k^*$ such that $y \in \mathcal{Y}_{k^*}$.

Therefore, for each $y \in \mathcal{Y}$, the corresponding ensemble output is:

$$Z_{\text{ens}} = (\perp, \ldots, \perp, y, \perp, \ldots, \perp),$$

where only the $k^*$-th component is equal to $y$, and all other components abstain.

This mapping from $y$ to $Z_{\text{ens}}$ is injective. Thus, the conditional entropy of $Y$ given $Z_{\text{ens}}$ is zero:

$$H(Y \mid Z_{\text{ens}}) = 0,$$

and mutual information is:

$$I(Z_{\text{ens}}; Y) = H(Y) - H(Y \mid Z_{\text{ens}}) = \log N.$$

*Interpretation.* Abstention removes the spurious random guesses that would otherwise obscure the label identity in the out-of-distribution case. Because each client specializes in a disjoint subset of labels, the ensemble output uniquely identifies the true label across the full label space. ∎

## A.2 THEOREM 2: OPTIMAL FUNCTIONAL AGGREGATION

Let assumptions (F1)–(F5) hold. Then the function $f^*(x)$ that minimizes the confidence-weighted alignment objective

$$\mathcal{L}(f^*) = \sum_{c=1}^{C} \mathbb{E}_{x \sim \mathcal{D}} \left[ \alpha_c(x) \cdot \|f^*(x) - f_c(x)\|^2 \right]$$

has the following closed-form solution:

$$f^*(x) = \frac{1}{\sum_{c=1}^{C} \alpha_c(x)} \sum_{c=1}^{C} \alpha_c(x) \cdot f_c(x).$$

ASSUMPTIONS (FUNCTIONAL AGGREGATION SETTING)

- **(F1) OSR-Aware Outputs:** Each local model $f_c(x) \in \Delta^{K+1}$ includes an abstention class $\perp$, and produces softmax outputs over $K + 1$ classes.
- **(F2) Confidence Weights:** For each input $x$, define confidence as $\alpha_c(x) = 1 - f_c(x)_\perp \in [0, 1]$. These weights are fixed and known at aggregation time.
- **(F3) Well-Defined Loss:** The local predictions $f_c(x) \in \mathbb{R}^K$ are bounded and measurable. The expectation over the data distribution $\mathcal{D}$ is finite.
- **(F4) Pointwise Alignment Loss:** The loss function is separable over inputs:

$$\mathcal{L}(f^*) = \mathbb{E}_{x \sim \mathcal{D}} \left[ \sum_{c=1}^{C} \alpha_c(x) \cdot \|f^*(x) - f_c(x)\|^2 \right].$$

- **(F5) Non-Degenerate Confidence:** For all $x$, at least one client has $\alpha_c(x) > 0$, ensuring the denominator in the closed-form expression is non-zero.

PROOF

Since the objective $\mathcal{L}(f^*)$ is a weighted sum of squared Euclidean distances, it is strictly convex, smooth, and differentiable in $f^*(x)$ for every input $x$. Moreover, by assumption (F5), the denominator $\sum_c \alpha_c(x)$ is non-zero for all $x$, ensuring a unique minimizer exists.

We may therefore apply the first-order optimality condition. For each input $x$, define:

$$J(f(x)) = \sum_{c=1}^{C} \alpha_c(x) \cdot \|f(x) - f_c(x)\|^2.$$

Taking the gradient with respect to $f(x)$ and setting it to zero:

$$\nabla J(f(x)) = 2 \sum_{c=1}^{C} \alpha_c(x)(f(x) - f_c(x)) = 0.$$

Solving gives the unique minimizer:

$$f^*(x) = \frac{1}{\sum_{c=1}^{C} \alpha_c(x)} \sum_{c=1}^{C} \alpha_c(x) \cdot f_c(x).$$

∎

## APPENDIX B: ADDITIONAL EXPERIMENTS

### B.1 CIFAR-10 PERFORMANCE COMPARISON

In this subsection we report results for additional federated learning algorithms beyond those included in the main results, focusing on methods often cited as effective under heterogeneous conditions. We use the CIFAR-10 dataset with 10 clients under extreme label skew as a stress test. As summarized in Table 1, most classical methods perform poorly in this setting, in many cases approaching random-guess accuracy. These findings reinforce the severity of heterogeneity as a fundamental obstacle and further motivate the development of approaches that can reliably operate under such conditions.

Table 1: Performance comparison of all methods on CIFAR-10 under extreme label skew

| Method | Accuracy (%) |
|---|---|
| FedAvg | 22.45 |
| SCAFFOLD | 21.71 |
| FedConcat | 18.75 |
| MoFedSAM | 25.83 |
| FedGF | 26.11 |
| FedGH | 11.41 |
| FedProx | 23.17 |
| FedMoon | 13.12 |
| FedDyn | 10.05 |
| FedSAM | 25.98 |
| FedVLS | 10.46 |
| FedOV | 43.27 |
| FedEOV* | 64.06 |

### B.2 HIGH CLIENT COUNT PERFORMANCE

To evaluate scalability beyond the main results, we extend the comparison to settings with 50 and 100 clients under extreme heterogeneity. As shown in Table 2, ensemble-based methods continue to perform well in these regimes, and the performance improvement of FedEOV* over FedOV is particularly pronounced as the client count grows. This highlights the robustness of ensemble-based approaches when information becomes increasingly fragmented. At the same time, we note that training in these high-client simulations is computationally costly, as the client processes are executed sequentially rather than in parallel, despite being inherently independent. This limitation affects only simulation efficiency, not the underlying scalability of the method.

Table 2: Performance comparison under high client counts

| Dataset | Clients | FedOV | FedEOV* | FedEOV-Pruned* | FedEOV-Distilled |
|---|---|---|---|---|---|
| CIFAR-100 | 50 | 78.68 | 92.23 | 72.96 | 47.90 |
| CIFAR-100 | 100 | 68.41 | 89.96 | 65.64 | 37.38 |
| Tiny-ImageNet | 50 | 79.03 | 89.42 | 48.00 | 24.62 |
| Tiny-ImageNet | 100 | 67.77 | 87.13 | 49.76 | 21.60 |

### B.3 ARCHITECTURE ABLATION

To examine whether the gains of FedEOV* arise consistently across architectures, we evaluate on larger and structurally diverse models, as shown in Table 3. All models are trained for 300 epochs, and as expected, accuracy improves further with additional training. However, this setting is sufficient to demonstrate the consistent advantage of FedEOV* over FedOV across architectures. Training larger networks is computationally intensive, since each client must be trained separately, making exhaustive exploration impractical. For this reason, we adopt a lightweight CNN in the main experiments to illustrate the trend, which is validated here by the consistent improvements observed on more complex architectures.

Table 3: Performance comparison across different model architectures on CIFAR-10

| Model | FedOV | FedEOV* |
|-------|-------|---------|
| AlexNet | 50.61 | 56.59 |
| All-CNN-C | 30.31 | 35.11 |
| Large-CNN | 41.26 | 48.88 |
| LeNet | 35.83 | 42.47 |

## B.4 FEATURE SKEW ANALYSIS

The benefit of ensemble-based methods extends beyond label skew, as we further illustrate in the feature-skew setting (Colored MNIST), with results summarized in Table 4. In this case, the label space is shared across clients, but each client observes a different domain (color variation). Under such conditions, ensemble methods remain advantageous: individual client models specialize to their domain, and at test time the OSR mechanism ensures that predictions are weighted toward the models most confident on a given input. At the same time, parameter-averaging methods also perform well in this scenario, since averaging reinforces commonalities across clients while mitigating noise, and no disjoint information is lost because all labels are present. Empirically, we observe exactly this trend, most methods achieve high accuracy under feature skew, reflecting that the setting is less challenging than extreme label skew.

Table 4: Performance comparison under feature skew (Colored MNIST)

| Clients | FedAvg | SCAFFOLD | MoFedSAM | FedGF | FedOV | FedEOV* | FedEOV-Pruned* | FedEOV-Distilled |
|---------|--------|----------|----------|-------|-------|---------|----------------|------------------|
| 5 | 94.66 | 94.66 | 96.52 | 96.87 | 98.70 | 98.66 | 98.32 | 98.65 |
| 10 | 94.73 | 94.72 | 91.14 | 93.05 | 94.07 | 94.09 | 97.86 | 98.36 |

## B.5 LABEL SKEW SENSITIVITY ANALYSIS

We also test under different levels of label skew using Dirichlet partitions (Table 5).

Table 5: Performance comparison under different levels of label skew (CIFAR-10)

| Dirichlet $\alpha$ | FedOV | FedEOV* |
|--------------------|-------|---------|
| 0.1 | 73.28 | 82.06 |
| 0.5 | 88.32 | 89.51 |
| 0.9 | 87.56 | 87.85 |

## B.6 MAIN RESULTS WITH ERROR BARS

To assess robustness, we report the mean and standard deviation across multiple random seeds for the key methods in the main results, as shown in Table 6. The deviations are generally small, confirming that the performance trends observed earlier are consistent and not driven by random fluctuations. This further strengthens the evidence for the reliability of FedEOV* across datasets and client settings.

Table 6: Performance comparison with standard deviation under extreme heterogeneity

| Clients | Dataset | FedOV | FedEOV* | FedEOV-Pruned* | FedEOV-Distilled |
|---------|---------|-------|---------|----------------|------------------|
| 5 | MNIST | $83.77 \pm 2.58$ | $87.62 \pm 1.96$ | $85.69 \pm 1.74$ | $67.20 \pm 3.32$ |
| | FMNIST | $68.89 \pm 3.92$ | $74.00 \pm 3.43$ | $75.41 \pm 2.43$ | $61.80 \pm 2.92$ |
| | SVHN | $51.50 \pm 8.42$ | $77.74 \pm 2.05$ | $75.62 \pm 1.03$ | $73.96 \pm 2.95$ |
| | CIFAR-10 | $69.09 \pm 2.00$ | $80.83 \pm 1.49$ | $68.68 \pm 0.16$ | $67.53 \pm 2.59$ |
| | CIFAR-100 | $86.14 \pm 1.15$ | $87.69 \pm 0.84$ | $67.16 \pm 0.59$ | $62.38 \pm 1.05$ |
| | Tiny-ImageNet | $65.76 \pm 2.07$ | $73.06 \pm 1.24$ | $40.86 \pm 1.28$ | $27.55 \pm 3.12$ |
| 10 | MNIST | $68.42 \pm 4.44$ | $85.61 \pm 1.20$ | $73.79 \pm 8.71$ | $45.72 \pm 2.64$ |
| | FMNIST | $64.64 \pm 2.07$ | $73.12 \pm 2.84$ | $65.22 \pm 11.36$ | $54.69 \pm 4.34$ |
| | SVHN | $37.29 \pm 10.43$ | $77.86 \pm 1.85$ | $74.34 \pm 0.34$ | $67.61 \pm 0.94$ |
| | CIFAR-10 | $43.27 \pm 1.93$ | $64.06 \pm 1.53$ | $49.59 \pm 2.75$ | $48.27 \pm 1.84$ |
| | CIFAR-100 | $77.07 \pm 1.13$ | $81.89 \pm 0.52$ | $54.58 \pm 0.14$ | $55.96 \pm 0.80$ |
| | Tiny-ImageNet | $73.09 \pm 5.54$ | $82.01 \pm 1.73$ | $32.87 \pm 0.23$ | $26.14 \pm 2.51$ |
| 20 | MNIST | $85.06 \pm 1.94$ | $89.69 \pm 2.84$ | $89.63 \pm 1.85$ | $74.59 \pm 0.80$ |
| | FMNIST | $71.50 \pm 3.41$ | $76.42 \pm 0.97$ | $79.19 \pm 0.59$ | $70.71 \pm 2.92$ |
| | SVHN | $63.24 \pm 2.26$ | $76.47 \pm 0.61$ | $73.46 \pm 0.91$ | $71.57 \pm 0.47$ |
| | CIFAR-10 | $62.77 \pm 0.79$ | $71.90 \pm 1.88$ | $58.93 \pm 0.67$ | $63.83 \pm 0.88$ |
| | CIFAR-100 | $85.40 \pm 0.70$ | $93.56 \pm 0.38$ | $55.50 \pm 0.44$ | $58.59 \pm 0.48$ |
| | Tiny-ImageNet | $64.56 \pm 3.62$ | $72.64 \pm 4.42$ | $26.13 \pm 2.36$ | $25.31 \pm 2.16$ |

## APPENDIX C: FURTHER THEORETICAL DISCUSSIONS

### C.1 COMPARISON OF ENSEMBLES AND PARAMETER AVERAGING BEYOND LABEL SKEW

Beyond heterogeneity due to label skew, federated learning systems face several practical challenges such as system heterogeneity, model heterogeneity, continual learning, and feature skew.

**System heterogeneity.** Clients often differ in compute power, communication availability, and participation frequency. Parameter averaging methods like FedAvg require synchronized updates, making them sensitive to such variation. In contrast, ensemble-based methods operate independently: each client updates its own model locally, with no need for synchronization. As a result, ensembles are naturally robust to system heterogeneity.

**Model heterogeneity.** In real-world deployments, clients may have different hardware capabilities or application needs that demand varying model architectures during training. FedAvg assumes identical architectures for parameter alignment, making it incompatible with such settings. Ensemble methods impose no such restriction, as aggregation occurs at the output level, allowing clients to train heterogeneous models without modification during local training.

**Continual learning.** Incorporating continual learning techniques into parameter-averaging frameworks is difficult due to their bilevel optimization structure and global synchronization. Ensemble methods, however, allow each client to update its local model independently using standard continual learning approaches. Clients can fine-tune models on new data and seamlessly update their contribution to the ensemble, enabling modular, scalable, and communication-efficient adaptation over time.

**Feature skew.** While our main paper focuses on label skew, ensemble-based aggregation with OSR is also effective when clients share labels but differ in input distributions. Parameter averaging can work in this setting by suppressing domain-specific noise and reinforcing shared features, but ensembles offer distinct advantages: (i) *domain specialization*, clients capture fine-grained cues from their own domain; (ii) *confidence-based routing*, OSR directs inputs to the most confident model, enabling mixture-of-experts behavior without extra rounds; (iii) *one-shot generalization*, diverse expertise improves robustness to unseen domains; and (iv) *permutation invariance*, output-level aggregation avoids neuron permutation and early-round misalignment. Thus, ensembles with OSR can match or exceed parameter averaging in feature-skew settings, especially under low-communication or dynamic conditions.

### C.2 INFERENCE LATENCY IN PRUNING AND DISTILLATION

Inference latency is a critical consideration for the practical adoption of federated ensemble methods, especially in edge and mobile deployments where compute resources are constrained. While accuracy improvements are important, real-world deployment often depends on whether a method can deliver predictions within strict time budgets. In this context, it is valuable to compare the theoretical latency of a *pruned ensemble* against a *distilled monolithic model*, as these two represent the primary deployment strategies for our proposed FedEOV framework.

We model inference latency in terms of FLOPs, throughput, and overheads. Let $F$ denote the number of floating point operations (FLOPs) for the baseline monolithic model, $\theta$ the sustained device throughput (FLOPs/s), and $\beta$ the constant launch and synchronization overhead. The latency of a single monolithic model is

$$\tau_{\text{mono}} \approx \beta + \frac{F}{\theta}.$$

For an ensemble of $C$ such models running across $n \leq C$ concurrent execution contexts, the latency is

$$\tau_{\text{ens, full}} \approx \left\lceil \frac{C}{n} \right\rceil \left( \beta + \frac{F}{\theta} \right).$$

We now introduce pruning. If a fraction $P$ of FLOPs is removed, then under the assumption of *linear FLOP reduction*, the remaining operations per branch are $(1 - P)F$. An efficiency factor $\alpha \in (0, 1]$ accounts for imperfect sparse kernel utilization ($\alpha = 1$ corresponds to ideal structured pruning). The

latency of a single pruned branch is

$$\tau_{\text{branch}} \approx \beta + \frac{(1-P)F}{\alpha\theta},$$

and the ensemble latency becomes

$$\tau_{\text{ens, pruned}} \approx \left\lceil \frac{C}{n} \right\rceil \left( \beta + \frac{(1-P)F}{\alpha\theta} \right).$$

Assuming negligible overhead ($\beta \approx 0$) and ideal efficiency ($\alpha = 1$), the condition for the pruned ensemble to be faster than the monolith is

$$\tau_{\text{ens, pruned}} \leq \tau_{\text{mono}} \quad \Longleftrightarrow \quad (1-P) \leq \frac{1}{\lceil C/n \rceil}.$$

This inequality highlights that higher pruning ratios are required when parallel execution is limited. Intuitively, if sufficient parallel contexts are available ($n = C$), then $\lceil C/n \rceil = 1$, and the ensemble is never slower than the single model, since no individual branch is larger than the monolith and all branches can be executed simultaneously. When parallelism is constrained ($n < C$), however, enough FLOPs must be pruned from each branch to compensate for sequential evaluation across groups of branches.

In practice, achieving such speedups requires industry-grade engineering. Our proof-of-concept experiments use standard PyTorch unstructured pruning, which zeros weights but does not reduce actual FLOPs in execution. Without specialized sparse kernels and hardware-optimized inference code, the wall-clock time remains dominated by dense operations. Furthermore, our experimental setup does not exploit inter-branch parallelism, so ensemble latency is higher than distilled latency in practice. The above derivation should therefore be interpreted as a theoretical upper bound, illustrating that under optimal parallelism and pruning conditions, pruned ensembles can match or surpass monolithic models in inference speed, but realizing these gains in deployment requires significant systems-level optimization.

## C.3 Theoretical Training Time Analysis: Pruning vs. Distillation

We first define the baseline training cost for the standard FedEOV setting without pruning or distillation. Let:

- $C$ = number of clients.
- $E$ = total local training epochs per client.
- $T_{\text{epoch}}$ = wall-clock time to train for one epoch on a given client model.

In one-shot FL, all clients train locally for $E$ epochs in parallel, so the total client-side time budget is:

$$T_{\text{FedEOV}} = E \cdot T_{\text{epoch}}.$$

**Pruning.** For FedEOV-Pruned, we use iterative pruning with short local training phases between pruning stages, followed by a final fine-tuning phase after the target sparsity is reached. Let:

- $S$ = number of pruning stages.
- $e_p$ = local epochs per pruning stage.
- $e_f$ = local epochs for final fine-tuning after reaching full pruning ratio.
- $T_{\text{prune}}$ = time to perform the pruning operation itself (negligible in practice).

The total client-side time is then:

$$T_{\text{pruned}} = S \cdot e_p \cdot T_{\text{epoch}} + e_f \cdot T_{\text{epoch}} + S \cdot T_{\text{prune}}.$$

In our practical schedule, we fix $E = S \cdot e_p + e_f$, so $T_{\text{pruned}} \approx T_{\text{FedEOV}}$ in addition to the small $S \cdot T_{\text{prune}}$ overhead. This ensures that pruning does not significantly increase client training time in the one-shot setting.

**Distillation.**  For distillation, each client first trains as in standard FedEOV, then the server performs additional training to transfer the ensemble's knowledge into a single student model. Let:

- $E_{\mathrm{KD}}$ = number of distillation epochs on the server.
- $T_{\mathrm{epoch}}^{\mathrm{server}}$ = time for one server epoch on the student model.

The total training time is:
$$T_{\mathrm{distill}} = T_{\mathrm{FedEOV}} + E_{\mathrm{KD}} \cdot T_{\mathrm{epoch}}^{\mathrm{server}}.$$
Since $E_{\mathrm{KD}}$ must be large enough to integrate knowledge from all $C$ client models into one network, this server-side cost is often substantial. In practice, initializing the student from an averaged client model does not help in the one-shot case, as our theoretical analysis shows that parameter averaging is ineffective under extreme label skew, and empirically this initialization offers no measurable gain.

**Comparative Analysis.**  If $E_{\mathrm{KD}} \cdot T_{\mathrm{epoch}}^{\mathrm{server}} \gg S \cdot T_{\mathrm{prune}}$, which is typical in realistic settings, then
$$T_{\mathrm{distill}} \gg T_{\mathrm{pruned}}.$$
That is, distillation almost always incurs a higher total training time than pruning in the one-shot FL setting, even before considering that pruning can be integrated into the fixed client budget.

**Data Requirements and Practical Considerations.**  Beyond raw training time, distillation in FL also raises the issue of server-side data. Effective distillation requires a representative dataset $D_{\mathrm{pub}}$ to query the ensemble and train the student. If such data is truly representative of all client domains, then the necessity of FL itself is questionable, as the central server already holds a high-quality dataset. If the public data is unrepresentative, the distilled model cannot capture knowledge from client distributions absent in $D_{\mathrm{pub}}$, a limitation widely discussed in the literature. While some hybrid FL variants assume public or synthetic data, our focus is the practical setting where no centralized dataset is available, in line with the core FL premise. In this setting, pruning provides a data-free compression path, avoiding the privacy and representativeness issues that make distillation controversial in federated learning.

C.4 Alignment Error in Parameter Averaging

In Theorem 3, we introduced an upper bound on the error of parameter averaging methods, which includes a term for *alignment error $\varepsilon_{\mathrm{align}}$*. This term captures the divergence between local model updates and the global objective, and has been the subject of extensive theoretical study in the federated learning literature. In this section, we trace the evolution of convergence analyses that attempt to characterize this alignment error, particularly under data heterogeneity.

Much of the early theoretical work on FedAvg focused on convex settings, where convergence is easier to guarantee Li et al. (2020c). These analyses established that parameter averaging can converge under standard assumptions, but they are far removed from the realities of deep learning and non-i.i.d. data. To relax these constraints, methods such as FEDPROX Li et al. (2020a) and FEDDANE Li et al. (2020b) introduced the notion of *bounded dissimilarity* between local objectives, formalized through bounds on gradient dissimilarity. This allowed for convergence guarantees in non-convex settings, but at the cost of a strong assumption: that clients are not too different. As highlighted in the follow-up analysis by Yuan & Li (2022), this assumption conflicts directly with the kinds of heterogeneity seen in real-world deployments, where client objectives can differ drastically due to skewed labels or non-overlapping distributions.

Other approaches, such as FEDNOVA Wang et al. (2020), addressed system heterogeneity by normalizing local updates based on computation effort, but still implicitly assumed aligned objectives. A more direct response to alignment issues came from methods like SCAFFOLD Karimireddy et al. (2020) and FEDDYN Acar et al. (2021), which explicitly model and correct *local drift* caused by data heterogeneity. SCAFFOLD reduces client drift using control variates that track update direction, while FEDDYN introduces a dynamic regularization term to adjust client loss functions toward global consistency. Both are motivated by the growing recognition that misalignment, if uncorrected, can severely degrade the performance of parameter averaging.

More recently, some analyses have considered settings where alignment can be recovered via structural assumptions like the *Polyak–Łojasiewicz (PL) condition*, which allows linear convergence under

non-convex but well-behaved landscapes Maralappanavar et al. (2023). It is well known that functions satisfying the PL condition share many of the favorable properties of strongly convex objectives despite not being strictly convex. However, these assumptions are of limited practical relevance in federated learning, where models are highly non-convex and data are non-i.i.d., so the PL-based bounds may not fully hold in realistic scenarios.

In summary, the alignment error $\varepsilon_{\text{align}}$ has received significant attention in the theoretical analysis of federated learning, with most convergence bounds expressed in terms of gradient variance, client dissimilarity, or structural conditions such as PL geometry. However, these bounds are typically derived under restrictive assumptions, such as convexity, bounded gradient dissimilarity, or well-behaved optimization landscapes, that often fail to hold in real-world federated settings characterized by deep models and highly heterogeneous, non-i.i.d. data. In contrast to this narrow focus on alignment, our formulation in Theorem 3 introduces a second, orthogonal source of error: *information collapse* due to disjoint label distributions. This provides a more complete picture of the limitations of parameter averaging under heterogeneity and motivates the need for alternative aggregation strategies.

### C.5 Positioning within One-Shot Federated Learning (OFL) Categories

Our work belongs to the class of ensemble-based one-shot federated learning (OFL) methods. However, ensembles represent only one branch of the broader OFL landscape. According to the summary in the recent survey Liu et al. (2025), existing methods can broadly be categorized into four types: parameter learning, knowledge distillation, generative models, and ensemble methods. Parameter learning methods, often inspired by FedAvg, operate directly on local parameters and seek statistical consensus; while simple, they are limited in handling heterogeneity and raise privacy concerns. Knowledge distillation methods instead transfer knowledge via compressed models or distilled data, and while they offer flexibility, they often lose fine-grained information during compression.

Of particular interest are generative approaches, which construct synthetic data that mimics the distributions of client datasets by leveraging techniques such as generative adversarial networks, variational autoencoders, or diffusion models. This strategy directly addresses data heterogeneity: once equipped with synthetic samples from all clients, the server can perform centralized training, thereby aggregating diverse client information into a unified global model. However, the very premise of this approach highlights its main weakness. Privacy is the ultimate constraint in federated learning, and transmitting any form of data to the server requires it to be simultaneously *fully informative* for effective learning and *fully secure* against leakage. Achieving this dual requirement in practice has proven extremely challenging. Synthetic data often fails to strike this balance, either sacrificing fidelity or compromising privacy. Moreover, these approaches impose significant additional training costs on the server. For these reasons, we instead advocate ensemble-based methods, combined with Open set-recognition (OSR), as a more practical way to preserve client knowledge while respecting privacy and maintaining efficiency.

## APPENDIX D: ALGORITHMIC FRAMEWORK

---

**Algorithm 1** FEDEOV - One Shot Federated Learning with Enhanced Open Set recognition

---

1: **Input:** Local model $f_i$, total epochs $E$
2: Initialize $f_i$
3: **for** training stage $s = 1$ to 3 **do**
4:     **if** $s == 1$ **then**
5:         Train $f_i$ for $E/3$ epochs using *augmented negative samples*
6:     **else if** $s == 2$ **then**
7:         Train $f_i$ for $E/3$ epochs using *adversarial FGSM samples*
8:     **else if** $s == 3$ **then**
9:         Train $f_i$ for $E/3$ epochs using *shuffled negative samples*
10:     **end if**
11: **end for**
12: **Server aggregates:** $f^*(x) = \frac{1}{\sum_i \alpha_i(x)} \sum_i \alpha_i(x) f_i(x)$
13: **Output:** Global model $f^*$

---

**Algorithm 2** FEDEOV-PRUNED – Pruning-Aware Ensemble Compression

---

1: **Input:** Local model $f_i$, pruning ratio $p$, pruning stages $P$, total epochs $E$, pruning stage length $e_p$
2: $E_{\text{rem}} \leftarrow E - P \cdot e_p$
3: Initialize $f_i$
4: **if** $p > 0$ **then**
5:     **for** pruning stage $k = 1$ to $P$ **do**
6:         **for** training stage $s = 1$ to 3 **do**
7:             **if** $s == 1$ **then**
8:                 Train $f_i$ for $e_{p1}$ epochs using *augmented negative samples*
9:             **else if** $s == 2$ **then**
10:                 Train $f_i$ for $e_{p2}$ epochs using *adversarial FGSM samples*
11:             **else if** $s == 3$ **then**
12:                 Train $f_i$ for $e_{p3}$ epochs using *shuffled negative samples*
13:             **end if**
14:         **end for**
15:         Prune lowest-magnitude weights to achieve $p$ (layerwise)
16:         Reinitialize remaining weights to $\theta_i^0$
17:     **end for**
18: **end if**
19: **for** training stage $s = 1$ to 3 **do**
20:     **if** $s == 1$ **then**
21:         Train $f_i$ for $E_{\text{rem}}/3$ epochs using *augmented negative samples*
22:     **else if** $s == 2$ **then**
23:         Train $f_i$ for $E_{\text{rem}}/3$ epochs using *adversarial FGSM samples*
24:     **else if** $s == 3$ **then**
25:         Train $f_i$ for $E_{\text{rem}}/3$ epochs using *shuffled negative samples*
26:     **end if**
27: **end for**
28: **Server aggregates:** $f^*(x) = \frac{1}{\sum_i \alpha_i(x)} \sum_i \alpha_i(x) f_i(x)$
29: **Output:** Global model $f^*$

---

## APPENDIX E: COMPUTE RESOURCES AND EXPERIMENTAL INFRASTRUCTURE

All experiments were run on a single workstation with the following specifications:

- **CPU:** 12th Gen Intel® Core™ i9–12900K
  - 16 physical cores (8 Performance-cores + 8 Efficient-cores), 24 hardware threads
  - Base clock 3.20 GHz, boost up to 5.2 GHz
- **GPU:** NVIDIA® GeForce RTX 4090
  - 24 GB dedicated GDDR6X VRAM
  - Driver version 32.0.15.6094 (released 14 Aug 2024)
- **System Memory:** 128 GB DDR5 RAM
- **Storage:** NVMe SSD (boot and dataset storage)
- **Software Environment:**
  - **Operating System:** Windows 11 Pro (64-bit)
  - **Python:** 3.11.8 (conda-forge)
  - **PyTorch:** 2.2.1+cu121
  - **CUDA Toolkit:** 12.1

## APPENDIX F: BROADER IMPACT

FL is increasingly critical in privacy-sensitive applications such as healthcare, finance, and mobile computing. Our method, FedEOV, addresses major limitations in FL by improving robustness to data heterogeneity and offering scalable one-shot ensemble aggregation.

By moving beyond parameter averaging, our approach allows for more personalized and effective client learning while minimizing server-side coordination. The pruning-based strategy further enables efficient deployment on resource-constrained devices.

These contributions make FL more accessible to real-world deployments, particularly in low-resource settings. However, care must be taken to ensure model fairness across underrepresented client groups. While our method reduces communication and computational costs, future work should explore fairness-aware aggregation and testing on more diverse data environments.