# OpenReview forum: "Rethinking Federated Aggregation Under Heterogeneity: Scalabe Ensembles With Open-Set Recognition"
_ICLR.cc/2026/Conference — ICLR 2026 Conference Withdrawn Submission_

### Official Review · Reviewer_UNjW · 2025-10-29

**Soundness:** 1
**Presentation:** 2
**Contribution:** 1
**Rating:** 0
**Confidence:** 4

**Summary:**

This paper aims at improving the federated learning with open-set recognition algorithm FedOV (Diao et al., 2023). The problem tackled here is to produce a good classification prediction in an FL context with extreme label heterogeneity, eg disjoint and 1 class per client.
The authors introduce FedEOV, a method similar to FedOV with an extra step introducing "shuffled-patch augmentations with smoothed transitions" to avoid learning spurious features of unknown classes.
They also provide theoretical justification of the open-set approach to tackle heterogeneity (in labels) in the FL context and test their method experimentally on standard classification datasets.
The authors also explore dimension reduction techniques like distillation and pruning to scale their method with the number of clients.

**Strengths:**

**Strengths**
- Good related work section and definition of open set voting in FL
- Reasonable experiments (standard datasets, several heterogeneity levels, extensions to tackle scalability issues)

**Weaknesses:**

**Weaknesses**
- Theorem 1 is correct, but not really informative: OSR is optimal because of assumption B4.
- Wrong theorem 2. Vector $x$ is not fixed in the problem (3) thus it cannot appear in the solution (4).
- No proof of theorem 3 in the appendix...
- There is a lack of clarity how the two paradigms FedAvg (and thus any standard FL algorithm) and FedOV (and thus FedEOV too) differ at training and at inference. This makes the discussion a bit confusing: in FL once the model is trained and aggregated one last time on the server, it's fixed. Then no other communication cost arises. As far as I understood, communication is still required for all clients to the server for every single inference, which typically makes it impractical in case of clients dropping the pool.

**Questions:**

Please find rapid comments and questions below.

**Comments**
- C1) the abstract and first 4 pages are too long. It would be better to focus on the differences between FedOV and FedEOV
- C2) (line 177 page 4) other papers tried to combine FL with pruning, especially in the case of model heterogeneity, which make sense in the context of ensemble learning as one can train locally "weak learner" and combine them on the server in a "strong learner". References:
    - Horvath et al, FjORD: Fair and Accurate Federated Learning under heterogeneous targets with Ordered Dropout (2021)
    - Alam et al, FedRolex: Model-Heterogeneous Federated Learning with Rolling Sub-Model Extraction (2022)
    - Yi et al, FedP3: Federated Personalized and Privacy-friendly Network Pruning under Model Heterogeneity (2024)
I let the authors check if the above references are relevant or not.
- C3) page 5, theorem 2, $f_c (x)_\bot$ not defined
- C4) a deeper explanation of the difference between FedOV and FedEOV is required here, especially experiments to check if FedEOV consistently brings an improvement. Or highlighting the spurious correlation learned by FedOV and not by FedEOV, visual examples are missing in this submission.
- C5) Figure 2 is not very clear, it's not sequential and present variants, which is confusing. What is the right-hand side part about? Maybe clearer to put the algorithm in the main text. Also, concerning the algorithm present in the appendix, I recommend a rewriting after drawing inspiration from FedOV algorithm that explicitly tells what is run on the clients and on the server.
- C6) Line 457-460 page 9, where is 16.76% visible? I disagree with the conclusion as it is written here. If FedEOV and FedOV perform the same in homogeneous setting, it implies the finale (3rd) stage of FedEOV dos not bring any benefit over FedOV. This does not prove that Open-Set Recognition is not effective. To conclude that, one need to compare to the baseline without OSR, eg FedAvg.

**Questions**
- Q1) in (4) there might be a mismatch in dimension as $f_c (x)$ do not necessarily belong to the same vector space for all $c$
- Q2) In the discussion after Theorem 2, it seems there is a comparison between FedOV with the assumption each $f_c (x)$ is optimal, ie trained, vs FedAvg training steps. Could you precise the discussion here to avoid confusion between training and inference stages. Also, aggregation in FL has never been a descent step.
- Q3) Is the experimental setting following the one of FedOV ? Why limiting to 20 clients and not 80 as authors do in the later? Are FedOV results reproduced?
- Q4) Why is the number of parameters increasing with the number of clients for open-set methods?
- Q5) What is "*" symbol for in the entire paper. It's always written "FedEOV*" not "FedEOV". Is there a footnote missing?
- Q6) Is $lr=0.001$ optimal for all methods?

---

> ### Author Response · Authors · 2025-11-17
> **Response to Reviewer Comments**
>
> We thank the reviewer for their comments. We will first address the weaknesses, then the questions, and finally the comments.
>
> **Comments on Weakness 1:**
> We do not agree that Theorem 1 is not informative. Theorem 1 is intended to establish that if abstention works correctly, then information collapse can be avoided. This is not automatic from the assumption of perfect OSR; the purpose of the theorem is to show explicitly that correct abstention preserves the full label information rather than suffering the collapse seen without OSR. Once this baseline is clear, the later results address the fact that OSR will be imperfect in practice, and that its errors become the limiting factor. This is also the motivation for focusing on improving the OSR component in our method.
>
> **Comments on Weakness 2:**
> The global model \(f\) is always a function of \(x\). If the concern is about notation, then we agree that omitting the explicit dependence on \(x\) in (3) made this a bit unclear, but this does not at all mean it is wrong. We will correct the notation so it is clear that the aggregation is performed at each input \(x\) and that the local models are fixed functions in this optimization.
>
> **Comments on Weakness 3:**
> The intention behind what we called Theorem 3 was simply to spell out the implication of the first two results and make the limiting role of abstention accuracy clear. It does not rely on additional assumptions and therefore does not require a separate proof. We now see that this would be more natural to present as an implication or corollary rather than as a full theorem, and we will revise it in that spirit.
>
> **Comments on Weakness 4:**
> We do not fully understand the concern. In our setup there is no communication at inference because each client receives the pruned ensemble and runs it locally. Communication happens only once during training for the final aggregation. After that, nothing more is required.
>
> ---
>
> ### **Q1 Response:**
> There is no dimension mismatch. All client models output over the same space: the full set of classes in the global system plus one additional token for the unknown class. Even though each client is trained only on its subset, the output layer dimensions are identical across clients, so the terms in (4) all lie in the same vector space.
>
> ### **Q2 Response:**
> We are not entirely sure we understand the exact point of your question, but if it is about whether this concerns training or inference, this discussion refers to the training stage, specifically the aggregation step after local updates.
>
> Regarding the quote “aggregation in FL has never been a decent step,” we believe you are referring to our line:
> **“FedAvg operates in parameter space and performs only an approximate gradient descent step.”**
>
> The standard FL objective is:
>
> $$
> L(w) = \frac{1}{N}\sum_{i=1}^N L_i(w)
> $$
>
> Performing gradient descent yields:
>
> $$
> w_{t+1} = w_t - \eta \frac{1}{N}\sum_{i=1}^N \nabla L_i(w_t)
> $$
>
> In the one-local-step setting (FedSGD), clients satisfy:
>
> $$
> \nabla L_i(w_t) = w_t - w_i
> $$
>
> Substituting and taking \( \eta = 1 \):
>
> $$
> w_{t+1}
> = w_t - \frac{1}{N}\sum_{i=1}^N (w_t - w_i)
> = \frac{1}{N}\sum_{i=1}^N w_i
> $$
>
> This corresponds exactly to FedSGD and is the true gradient step.
> FedAvg becomes approximate only because clients take multiple local steps before sending updates, so \(w_i\) is no longer a one-step descent update from \(w_t\).
>
> This approximation view is standard in the FL literature, so we did not expand it in the main text.
>
>
> ### **Q3 Response:**
> Our focus is specifically on the fully disjoint label setting, so the exact experimental setup is not the same as in FedOV, though it is similar. We use the authors’ implementation directly and run it within our pipeline. FedOV also includes some experiments with disjoint labels, although their partitioning procedure differs slightly from ours. The only directly comparable case is the 10-client, one-class-per-client setting, and the reproduced results in that configuration are quite similar.
>
> We also include results with more than 20 clients for larger datasets in Appendix B2. Running all datasets with higher client counts for every baseline and across multiple seeds was not feasible due to the computational cost. Increasing the number of clients significantly increases training time, especially for round-based baselines requiring repeated communication rounds.
>
> ### **Q4 Response:**
> The parameter count increases with the number of clients because the open-set methods we study are ensemble-based, so the aggregated model includes one classifier per client. This scalability issue is the central premise of the paper and is the motivation for evaluating pruning to reduce the final model size.
>
> ### **Q5 Response:**
> We are not entirely sure we understand this point. Could you indicate where the method name appears in a different style or with an unintended symbol so we can correct it?

---

> > ### Author Response · Authors · 2025-11-17
> >
> > ### **Q6 Response:**
> > The local learning rate concerns only the client-side training and is not relevant to the differences between the aggregation methods. While the local objectives can differ slightly across methods, the overall local optimization procedure is not significantly different. For this reason, we consider the chosen learning rate equally appropriate for all methods.
> >
> >
> > ---
> >
> > ### **Response to Comments (C1–C6):**
> > We thank the reviewer for these helpful suggestions. We will shorten the opening section, add relevant pruning citations, fix the missing definition, clarify FedOV vs. FedEOV, revise the figure and algorithm for clarity, and update the homogeneous-setting interpretation.
> >
> > For **C6**, the 16.76% value comes from summing accuracies across datasets and client settings for FedOV and FedEOV in our extreme heterogeneity setup and computing the relative gain. We will clarify this in the revision. Our point about homogeneous performance was simply that stronger OSR does not help when data are already aligned, which is consistent with prior work. Since this is not an important conclusion, we will remove it to avoid confusion.

---

> > ### Comment · Reviewer_UNjW · 2025-11-18
> > **Answer to comment on Weakness 1**
> >
> > I understand authors perspective on this point. That being said, I believe the way it is formulated should be changed to clearly expressed it is about an idealized case, maybe one could call considered model an "OSR Oracle" rather than "model trained on disjoint labels with abstention".

---

> > ### Comment · Reviewer_UNjW · 2025-11-18
> > **Answer to comment on Weakness 3**
> >
> > I understand the authors intention, Yet, even a corollary should be proved. In Theorem 3, several quantities are not even defined.

---

> ### Comment · Reviewer_UNjW · 2025-11-18
> **Comment on Weakness 2**
>
> I thank the authors for taking into account my review and for their answers.
>
> I think my concern about Theorem 2 has not been understood. In (3), the considered objective is independent in $x$ as it is an expectation over the distribution $x \sim \mathcal{D}$, so it's mathematically incorrect that (4) is the solution as it is depending on a specific $x$, which is not defined.

---

> ### Comment · Reviewer_UNjW · 2025-11-18
> **Answer to comment on Weakness 4**
>
> Thanks to the authors' comment and an additional reading of Section 3.3 of (Diao et al, 2023) to understand the setting (this relates to my comment C5 about the lack of clarity of Algorithm 1 compared to the one in FedOV paper). I acknowledge that I did not understand precisely the training and inference stages.
>
> Could you please confirm my understanding?
> - At training time, models are trained locally only. At the end of the training, each client's model is shared with the server and then all models (the ensemble), C-1 models, are shared back to each device.
> - At inference, ensemble prediction is applied locally on each client, ie, C model inferences
>
> If this is the case, then
> - There is an obvious privacy issue resulting in sharing models between clients
> - Inference computation cost is proportional to C on each device which prevents the usage of this technique on real-world scenario where edge devices are limited in memory and compute

---

> ### Comment · Reviewer_UNjW · 2025-11-18
> **Response to answers to questions**
>
> - Q1) I understand better now, thank you. This should be explicitly written in the definition of $f_c$ because it implies all clients are aware of all existing classes, which is an important hypothesis.
> - Q2) I wrote my comment to rapidly and I acknowledge it's definitely a gradient step over the overall objective for FedSGD.
> - Q5) I miss typed and meant * symbol. But now I guess that you wanted to highlight your method. I was just confused as I thought it was a footnote symbol.

---

### Official Review · Reviewer_NS4K · 2025-10-29

**Soundness:** 2
**Presentation:** 2
**Contribution:** 1
**Rating:** 4
**Confidence:** 4

**Summary:**

The paper proposes an algorithm to address extreme label heterogeneous data in FL setting (where the clients have a disjoint set of classes). Each client learns a classifier using its own local data and prunes the model as it does. Then shares the model with the server. Each of these models is trained to differentiate between the samples from classes that are not there in the local data. This is done by training with an extra 'unknown' class. The method augments the local samples to produce negative samples to train the 'unknown' class. The server uses the ensemble of (pruned) client models for inference.

**Strengths:**

1. Communication efficiency in FL is an important problem.
2. The paper explores an interesting solution using ensembles and open set recognition (if the category of a test sample was seen during training)

**Weaknesses:**

1. Authors should improve readability by breaking long paragraphs at logical places.
2. I'm unclear about the setting. Is each model stored on every device and used for inference? Or does the server store all the models and make the inference?
3. I think the assumptions in the appendix should be in the main document. And the theorems need to be formulated taking them into account. Otherwise, the current state of the theorems is confusing. For example, in Theorem 1 statement, it is unclear what is Z_nosr and Z_osr and a natural question is how the OSR loss function is defined.
4. The novelty of the paper is a bit concerning. The authors mentioned three contributions: i) theoretical analysis, ii) augmentation to generate negative samples, and iii) pruning models.
	a. I feel the theoretical part needs to be more formal and concrete by bringing in material from the appendix.
	b. The second contribution is explained as a single line 331 by only saying "shuffled patch augmentations with smoothed transitions". It is too short to be a main contribution. For example, why does this augmentation help the model learn to differentiate between seen and unseen classes?
	c. The pruning module uses existing methods in pruning.
5. The paper needs to cite previous works that it got inspiration from. For example, pruning methods.
2. Explain the pruning method used in line 342. The details are unclear - how to decide the threshold for pruning. Also, I'm unsure about: "Surviving weights are reset to their original pre-training values before training resumes" (line 344). Can you elaborate more on why this will work? The main challenge in FL + pruning comes from parameter averaging of pruned models, which is not relevant in this context.
6. It is unclear how the ensemble predictions are combined to produce the final prediction, taking into account the OSR confidence values.
7. I'm not sure why the authors have made the case using on-shot FL, whereas the method seems to be one-shot.


Detailed comments:
1. fix citation to remove redundant string: Guha et al. Guha et al. (2019) in line 165
2. The main paper should include the details for the experiment settings regarding datasets, models, and FL setting.
3. One of the major claims of the paper is that the resulting method is efficient, but it only shows the parameter count. It will be more convincing to talk in terms of CPU cycles, peak memory usage, and GPU memory.
4. None of the results have confidence bounds.

**Questions:**

As above.

---

> ### Author Response · Authors · 2025-11-17
> **Response to Reviewer Comments**
>
> We thank the reviewer for their comments. We will first address the weaknesses and then the detailed comments.
>
> **Comment on Clarity (W1):**
> We appreciate the pointer and will break the longer paragraphs at appropriate places in the revision.
>
> **Clarification of the inference setup (W2):**
> Once the pruned ensemble is constructed on the server, the complete ensemble is sent back to each client. This ensures that inference requires no further communication with the server; every client can run inference locally using the full ensemble without relying on repeated queries. During training as well, there is no communication beyond the single upload of the locally trained (and pruned) model, since each client trains solely on its own data and aggregation happens only once through ensembling on the server. This eliminates any iterative parameter exchange and makes the entire procedure single-round in training and communication-free during inference.
>
> **Comment on theorem assumptions (W3):**
> As suggested, we will work to include the assumptions for the proofs in the main text, although space constraints may make this challenging.
>
> **Comment on Contribution concerns (W4):**
> Our intention was to build a unified case around a single argument: that parameter averaging is fundamentally ill-equipped to handle the core issues introduced by extreme heterogeneity. To us, this is the primary contribution, because it challenges the dominant paradigm in FL and makes clear why an ensemble-with-OSR approach is more aligned with the structure of the problem. The theoretical analysis motivates this shift by showing that ensemble performance is bottlenecked by OSR quality, which is why we focus on improving OSR. The pruning component then addresses the main practical limitation of ensembles, allowing them to remain lightweight at inference. The novelty here is not in the pruning mechanism itself, but in advocating pruning as a lightweight alternative to distillation for reducing ensemble size in this setting. In this sense, the contribution should be viewed holistically, as an argument in favor of shifting research focus toward approaches beyond parameter averaging, given its inherent limits in addressing heterogeneity and communication constraints.
>
> **Comment on pruning citations (W5):**
> Thank you for the suggestion. We will cite the relevant pruning papers accordingly.
>
> **Comment on the pruning method (W6):**
> The pruning step follows the well-known Lottery Ticket Hypothesis / iterative magnitude pruning procedure. It is applied entirely locally, with no parameter averaging, so each client simply trains its own sparse subnetwork. We agree this should have been cited explicitly and will add the appropriate references.
>
> **Comment on ensemble inference (W7):**
> The ensemble prediction is an average of the models, weighted by their OSR confidence. Alternatively, following FedOV, one may use the top-k most confident models. We will clarify this and may bring the pseudocode into the main text if space allows.
>
> **Regarding the “on-shot FL” (W8):**
> Could you clarify what you meant by “on-shot” in this context, and whether it refers to a specific FL term or to a possible typo or miscommunication on our side?
>
> ---
>
> ### **Detailed Comments**
>
> **Typo at line 165 (comment 1):**
> Thank you for noting this. We will correct the typo.
>
> **Clarification on the experimental setting (comment 2):**
> The details of the datasets, model architectures, and the federated setup are included in the main paper’s experimental section.
>
> **Clarification on efficiency (comment 3):**
> Appendix C.2 and C.3 discuss inference cost, training cost, and scalability considerations of the ensemble approach beyond the parameter count.
>
> **Clarification on confidence bounds (comment 4):**
> Appendix B.6 contains confidence bounds for the key methods.

---

### Official Review · Reviewer_LJE4 · 2025-10-31

**Soundness:** 3
**Presentation:** 3
**Contribution:** 3
**Rating:** 6
**Confidence:** 3

**Summary:**

This paper critiques the dominant parameter-averaging paradigm in FL, arguing that it is fundamentally flawed for settings with high statistical heterogeneity, particularly extreme label skew. Averaging-based methods may suffer from information loss and local drift, which cannot be adequately corrected. As an alternative, the paper champions an ensemble-based approach using OSR, which aggregates models in the function space, thereby preserving client-specific knowledge. The proposed methods include FedEOV, an enhanced version of the SOTA ensemble method, FedOV, using a more robust three-stage negative sample generation, and FedEOV-Pruned, which addresses the scalability limitation of ensemble methods.

**Strengths:**

- The core argument that parameter averaging is the wrong approach for heterogeneity and ensemble-based OSR is clear, direct, and well-motivated. It seems reasonable to me.

- Theoretical grounding is provided for the claims made.

- The empirical results look strong, demonstrating significant improvement.

**Weaknesses:**

- The paper dismisses the poor performance of the hybrid method FedConcat, attributing it to hyperparameter sensitivity. While this may be true, it slightly weakens the comparative analysis. A more in-depth exploration of why its clustering mechanism fails under extreme skew would be more conclusive than a note on hyperparameters.

- The 3-stage OSR training for FedEOV is a key contribution. However, there is no ablation study to show the marginal benefit of each stage. Is the performance gain primarily from Stage 3?

- In the main experiments, all models use a simple CNN with two convolutional layers and one fully connected layer. While the appendix mentions experiments with larger models, the core claims in the main paper would be strengthened if some of them could be included in the main body.

**Questions:**

- FedEOV-Pruned is described as using iterative pruning with reinitialization. How crucial are these specific (and more complex) "lottery ticket" style elements? How does it compare to a simpler, one-shot pruning of weights based on magnitude at the end of local training? Is the iterative process essential for maintaining accuracy at high sparsity?

- The paper focuses on the extreme skew case, so how does the performance gap between FedEOV and parameter-averaging methods (like FedGF) change as heterogeneity decreases?

- Also see weaknesses please.

---

> ### Author Response · Authors · 2025-11-17
> **Response to Reviewer Comments**
>
> Thank you for the comments and constructive feedback. We will address the questions first, followed by the weaknesses.
>
> **Q1 Response:**
> The iterative pruning with reinitialization does matter. As Figure 3 shows, one-shot pruning works at lower sparsity but drops off substantially once the pruning level becomes high, whereas the iterative lottery-ticket style approach remains much more stable. This becomes important when all pruned client models are combined into one ensemble. Looking back, we realize we did not spell out this difference clearly in the text, and we will add a short explanation in the revision.
>
> **Q2 Response:**
> We did include results for Dirichlet–0.1 and the homogeneous setting for FedAvg, which is the main parameter-averaging baseline. We did not extend this to more recent averaging methods because our focus is specifically on the high-heterogeneity regime. As discussed earlier in the paper, parameter averaging performs reasonably well in homogeneous or mildly heterogeneous settings, and our key argument concerns the core limitations introduced by heterogeneity, limitations that become fully exposed only in the extreme case. For this reason, the most detailed comparison is concentrated on the extreme-heterogeneity scenario.
>
> **W1 Response:**
> We appreciate the point. We will examine FedConcat’s failure mode in more detail. While hyperparameter sensitivity appears to be part of the issue, we agree that a deeper explanation would strengthen the comparison.
>
> **W2 Response:**
> Since Stage 3 is the only part that differs from FedOV, it is naturally where the performance gain comes from. The harder, smoothed negative samples introduced there prevent the shortcut cues learned in the original method.
>
> **W3 Response:**
> We used the small CNN in the main experiments because the full set of evaluation combinations is extremely large. Each method must be run across multiple datasets, multiple heterogeneity settings, multiple client counts, multiple seeds, and, for averaging-based methods, many communication rounds. In our preliminary tests, the small CNN reached strong performance on all datasets with very few epochs, while larger models required significantly more training time to reach higher accuracy. For this reason, the larger-model experiments were limited to a single dataset and single seed and placed in the appendix. Given space constraints, it did not seem worthwhile to move them into the main paper.

---

### Official Review · Reviewer_rKiB · 2025-11-03

**Soundness:** 1
**Presentation:** 2
**Contribution:** 1
**Rating:** 2
**Confidence:** 4

**Summary:**

This paper presents FedEOV, an ensemble-based algorithm for one-shot FL designed for scenarios of extreme label skew where the clients own disjoint label sets. The authors argue that when the clients own and train on completely different labels, parameter averaging (e.g. FedAvg) is bound to fail due to information collapse and misalignment. Instead they propose training with open-set recognition (OSR) like in FedOV, and further introduce model pruning to compress the resulting ensemble. The paper includes some theoretical analysis on mutual information, functional aggregation, and the bounds of averaging vs ensembling error (but see W2 below). Experiments cover MNIST, FashionMNIST, SVHN, CIFAR10/100, and TinyImageNet and multiple baselines in 3 heterogeneity settings, with the proposed method coming out on top (but see W4).

**Strengths:**

The paper offers clear problem framing, despite dealing with an extreme edge case and requiring very strong assumptions. The experimental coverage is comprehensive encompassing multiple datasets and baselines. The limitations of previous work it attempts to address, namely the non scalability of ensembles, are very relevant and the proposed approach via pruning seems intuitive.

**Weaknesses:**

## 1. Limited applicability of the motivating scenario

One-shot FL can address important challenges in resource-constrained environments and large-scale model deployments. However, this paper's focus is on image classification (on toy datasets), with a 3 layer CNN, on completely disjoint label distributions, representing an extreme edge case. To strengthen the motivation, I encourage the authors to: (a) provide concrete real-world examples where image classification tasks would exhibit disjoint label sets across clients, and (b) discuss whether the theory generalizes to more common scenarios with partial label overlap. The information collapse argument would also benefit from extending beyond the single-class-per-client example to demonstrate robustness across varying degrees of label skew.

## 2. Theoretical framework requires rehauling

* In Theorem 1, two of the assumptions are mutually inconsistent A3: Deterministic Prediction on In-Distribution Labels means the network is perfect but A.4 Uniform Prediction on Out-of-Distribution Labels (No OSR) requires complete randomness in misclassification of unseen labels. For instance, if a model perfectly classifies ships and dogs (A3), it seems unlikely to uniformly misclassify airplanes across all known classes (A4), rather, airplanes would likely be consistently classified as ships due to visual similarity.
* Theorem 2: the optimality claim would be more compelling if the authors justified why the objective in Eq. (3) is the appropriate choice for federated aggregation. Currently, the theorem shows that ensembles minimize a specific objective, but doesn't establish why this objective is superior to alternatives. Connecting this to a global federated learning objective or comparing different functional aggregation objectives would strengthen this result.
* Theorem 3 is not currently contributing to the paper's argument, since it does not compare the averaging error and the ensembling error. The inequality $\mathcal{E_avg}$ > $\mathcal{E}_{ens}$ is qualitative, not predictive. Consider either deriving bounds under some specific conditions or repositioning this as an empirical observation rather than a theoretical result.

## 3. Critical evaluation methodology issue

The results presented in Tables 2, 3, 4 caught my attention: Tables 3 (Dirichlet 0.1) and 4 (IID partitions) are easy to cross reference with other literature, whereby it becomes evident the proposed method has extremely high results, e.g. FedEOV - 20 clients - CIFAR100 achieves **98.23%** with a 3-layer CNN on CIFAR100.

After reviewing the provided code, I have a significant concern with the experimental protocol. The authors do not form disjoint label sets only for the training sets, **but also the test sets**. The ```compute_ensemble_accuracy``` function then iterates over client-specific test sets (test_loaders), where each test set appears to be partitioned similarly to the training data. For extreme label skew, this means each client's model may be evaluated primarily on classes it was trained on, rather than on a global test set representing the full label distribution. This could substantially inflate reported accuracies and explain results that appear inconsistent with other benchmarks.

Currently the partitioning strategy means in many of the experimental scenarios every client has a separate test set with only 1 class, and the task for the ensemble training is binary classification (class image vs abstention token), which of course can be done nearly perfectly, while the models trained via averaging receive conflicting gradient updates that do not allow them to make meaningful progress in this task.

## 4. Incremental novelty and clarification of technical contributions needed

The paper's main idea is the same as FedOV, so it would benefit from more clearly delineating its contributions relative to FedOV. Moving the core algorithm from the appendix to the main text with a clear walkthrough would help to this effect, along with expanding the technical description of the enhanced OSR mechanism and pruning strategy which is currently relegated to two paragraphs on page 7/9. Additionally, while the enhanced negative sample generation and pruning strategies are interesting, additional discussion would help readers understand: why harder negative samples are necessary given the relatively simple datasets used, and how the pruning insights on this 3 layer CNN would transfer to modern architectures (ResNets, Vision Transformers).

**Questions:**

1. Please clarify the evaluation protocol. Specifically: (i) Are test sets partitioned identically to training sets? (ii) Is the final accuracy computed over a single global test set or averaged across client-specific test sets? (iii) Can you provide results using a standard, unpartitioned test set for comparison?

**Details Of Ethics Concerns:**

The code is provided via github link from an anonymous profile, rather than an anonymous platform like https://anonymous.4open.science/ . The repo itself has 2 stars from other profiles potentially leaking information about the authors' identity.

---

> ### Author Response · Authors · 2025-11-17
> **Response to Reviewer Comments**
>
> We are grateful for the reviewer’s time and for the depth and rigor with which they engaged with our submission. We tried to address the main points briefly at this stage and hoped to expand on specific issues in the follow-up responses, but given the level of detail in the critique, to adequately address each point, some of our responses ended up longer than we initially intended.
>
> **Short Clarification on the anonymity flag:**
> We believe the repository complies with anonymity guidelines, as the account is private and contains no identifying information, email, or external links. As the repository is publicly accessible, public interactions such as stars or forks are not indicative of authorship and therefore do not violate anonymity. We do appreciate the reviewer’s suggestion and acknowledge that the platform they mentioned offers a more robust approach to maintaining anonymity, which we will adopt in future submissions.
>
> ---
>
> ### **Comments on limited applicability (W1):**
> We would like to begin by addressing the point on limited applicability of the extreme label skew scenario. The motivation of our setting was not to reproduce real-world deployment scenarios but to deliberately isolate and make explicit the core limitations introduced by heterogeneity. The fully disjoint label case serves as a controlled stress test that exposes the core failure modes of traditional federated methods (aggregation misalignment and information loss). That is not to say there could be no real-world cases resembling this setup. For instance, specialty clinics such as cardiology, dermatology, and ophthalmology label different conditions entirely, effectively creating disjoint label spaces under privacy constraints.
>
> Also the argument is not limited to the “single-class-per-client” case. Theorem 1 is expressed in terms of M, the number of classes per client. If what you meant was to extend this to the overlapping, Dirichlet-induced heterogeneity setting, the mathematics for that case would become considerably more obscure and less interpretable. This is, in fact, the core motivation for our chosen scenario: by pushing the heterogeneity to the extreme, the information collapse problem becomes analytically transparent, whereas in the overlapping label case it is more difficult to formalize and perhaps also to see intuitively.
>
> Although providing a clean theoretical generalization to the overlapping-label scenario is difficult, we agree that adding a short discussion explaining why the same failure mode persists there would strengthen the connection to more common FL settings.
>
> ---
>
> ### **Comments on Theoretical concerns (W2):**
>
> **Theorem 1:**
> Regarding the concern that A3 and A4 may be inconsistent: if simplified, A3 assumes perfect deterministic accuracy on known labels, while A4 assumes randomness on unknown labels. We do not view this as a contradiction, since perfect knowledge of the known domain does not necessarily imply certainty about behavior in the unknown domain.
>
> Your example is well taken: some unseen classes will indeed be semantically closer to certain seen classes, and the model might consistently predict those. However, since this structure varies across datasets and cannot be modelled universally, a uniform random assumption follows naturally from the principle of insufficient reason. In other words, we can model our lack of knowledge of these semantic relationships uniformly. This simplification captures the intended behavior without affecting the substance of the result. Whether the model misclassifies an unknown input consistently to one class or uniformly at random, the key point remains that without an open-set mechanism, the model is forced to assign every OOD input to one of its known labels, thereby degrading the ensemble’s prediction.
>
> **Theorem 2:**
> A deeper exposition of why this objective captures the challenge of transferring each local model’s domain-specific information into the global model would be valuable. We recognize that this connection warrants a clearer explanation.
>
> **Theorem 3:**
> In retrospect, we see your point regarding what we had presented as Theorem 3. Its role in the paper is not to introduce a new theoretical result, but to convey what follows naturally from the first two: that the ensemble error is ultimately constrained only by the quality of open-set recognition, whereas parameter averaging remains constrained by more fundamental issues. This observation is what guided our focus on strengthening the OSR component, which is reflected in the improvements achieved by FedEOV. Thinking about it now, this is perhaps better expressed as an implication of the earlier theorems rather than as a standalone theorem.

---

> > ### Comment · Reviewer_rKiB · 2025-11-18
> > **Theory**
> >
> > Re: theorem 1, A4 requires *absolutely no knowledge* gained by the known classes to be useful for classifying the unknown classes, so while I agree "perfect knowledge of the known domain does not necessarily imply certainty about behavior in the unknown domain" this is not sufficient for A3 and A4 to not be contradictory, as even a modicum of transferable knowledge would break A4. Transferable & knowledge being used here in a hand-wavy manner for brevity, but I do hope the point comes across. Since the authors put this claim forth as a Theorem, and the first one at that, it needs to be discussed as such and treated with at least some rigor (I'm not diving into details here, this is obviously contradictory), so while the overall intuition might be correct, the fact remains this theorem is wrong.
> >
> > I appreciate how gracefully the authors have taken the feedback re: Theorems 2 and 3 which was absolutely meant to be constructive and I do hope this discussion results in a strengthened theoretical section in future iterations of the manuscript.

---

> > ### Comment · Reviewer_rKiB · 2025-11-18
> > **Novelty & Applicability**
> >
> > - My comment about the "single class per client" was specific to the information collapse argument, which the authors themselves frame as such on line 233, I, of course, understand other parts of the manuscript do not assume this.
> >
> > - Regarding heterogeneity, this is a sore point in FL literature in general, as research practice where heterogeneity is put forth as singularly important comes at odds with practice (see https://www.youtube.com/watch?v=Ey-lO4XKmgg and https://arxiv.org/abs/2510.12595 if interested in supporting evidence). A Dirichlet non-IID split with a low alpha already requires a lot of suspension of disbelief regarding its applicability to real scenarios, but the proposed setting here requires a lot more. For example, in the proposed scenario of "cardiology, dermatology, and ophthalmology" clinics working together, what are we even trying to achieve? Why would we engage these clinics in collaborative model training to begin with? They do not even work with the same type of image (cxr/ct/mri vs dermoscopy/photo vs retinal scan).
> >
> > - Regardless of the above point, my understanding of the logic put forth in the rebuttal is: non-IID data -> label skew -> extreme label skew (<- all analysis is done here). Every arrow tackles only a narrow subset and I did not see any claims or arguments as to why/how an analysis performed on this narrow subset holds for the superset. So even if the statement "by pushing the heterogeneity to the extreme, the information collapse problem becomes analytically transparent" were to be true, there is still no motivation to do this. There needs to be an argument to connect this special case to the general problem.
> >
> > - The bar for novelty cannot be lowered simply because the problem is under-researched in the opinion of the authors. If the point of the manuscript is not to propose a method, in a well-motivated setting, sufficiently explained, that clearly goes beyond previous methods in a non-trivial way, but rather to serve as a position paper that promotes this type of one-shot FL and provides a couple of incremental improvements on an existing method, then this might be an interesting paper but not one that I would expect to see accepted at ICLR.

---

> > > ### Author Response · Authors · 2025-11-19
> > > **Gratitude for the Reviewer’s Detailed Assessment**
> > >
> > > The paper you mentioned, highlighting how academic FL research may focus on problems that differ from those encountered in practice, is indeed very interesting. We will look into this direction and consider how it may inform a revised motivation in future iterations.
> > >
> > > We also wanted to briefly clarify the issue behind the inflated results. While the evaluation logic itself was correct, the dataset construction inherited from the codebase we built on had an unintentional data-handling issue that caused a subtle train/test leakage and consequently inflated the reported performance of all algorithms. Although our preliminary corrected runs still suggest an advantage for ensemble-based approaches, for our method over FedOV, and for pruning, the full set of experiments is too large to recompute within the discussion phase. Given the scope of this rework, we feel it is most responsible to withdraw the manuscript. Thank you for the rigorous evaluation and for pointing out the inconsistency.

---

> ### Author Response · Authors · 2025-11-17
> **Weakness 3 response**
>
> ### **Comments on the evaluation protocol issue (W3):**
>
> **Q1. Are test sets partitioned identically to training sets?**
> Yes. Each client’s test loader contains only the classes assigned to that client, following the same partitioning strategy as the training data.
>
> **Q2. Is the final accuracy computed over a single global test set or averaged across client-specific test sets?**
>
> It is neither a single global test set in the literal sense, nor an average over clients. There seems to be some misunderstanding about the evaluation function, so we first address what we believe is causing the confusion.
> You seem to assume that each client receives a test set with only one class and that, during evaluation, the model performs a binary task (class vs abstention) at each client. This is not what happens. The abstention task (which by the way is a K+1 class classification not binary, K being the number of classes per client) is only part of the training stage. At test time, the ensemble performs an N-class classification task, where N is the total number of classes in the dataset. For CIFAR-100 this means the ensemble outputs a 100-dimensional vector for every test example, and there is no abstention or “unknown” class present in the test output.
>
> During evaluation, we simply go through each client’s test loader, count how many total samples we saw, how many predictions were correct, and then compute one accuracy as total_correct divided by total_samples. This is exactly equivalent to taking all the clients’ test data, concatenating it into one global test set, and evaluating once on that. The fact that we loop over clients is only an implementation detail.
>
> Here is a step-by-step walk through of the `compute_ensemble_accuracy` function you mentioned, showing these points clearly.
>
> The evaluation goes through each client’s test loader:
> `for loader_id, dataloader in enumerate(test_loaders):`
>
> For each batch, we run ensemble inference. The important line is:
> `output_vector = ensemble_inference(X, contrastive_classifiers, aggregation_method=aggregation_method, verbose=verbose)`
>
> Inside ensemble inference, each discriminator outputs a softmax. The abstention dimension is removed by taking only `softmax_output[:, 1:]`, so the returned tensor never contains an abstention class. Some key lines:
> `softmax_output = torch.softmax(output, dim=1)`
> `output_vector.append(softmax_output[:, 1:])`
>
> These per-discriminator class probabilities are stacked and aggregated into a final probability vector over the actual dataset classes only. The ensemble output is therefore always a full N-dimensional class vector, not a binary vector.
>
> The prediction is obtained using:
> `predicted_class = output_vector.argmax(dim=1)`
>
> This argmax is over the full class space (for example, 100 classes in CIFAR-100). There is no abstention or binary decision at test time.
>
> Correct and total are accumulated globally, not averaged per client:
> `correct += (predicted_class == target).sum().item()`
> `total += target.size(0)`
> `total_correct += correct`
> `total_samples += total`
>
> At the end, a single accuracy is computed using all test samples from all clients:
> `overall_accuracy = total_correct / total_samples * 100`
>
>
> **Q3. Can you provide results using a standard, unpartitioned global test set?**
> Yes, we can, and we will include these results in our subsequent reply. Although the evaluation function appears correct, we will double-check the data-class generation to ensure there is no error or train leakage.
>
> **W3 (CIFAR-100 performance):**
> Regarding the surprisingly high CIFAR-100 performance, we note that both FedOV and FedEOV aggregate ensembles whose parameter count scales linearly with the number of clients. Table 1 illustrates how ensemble size grows with client count, explaining the increased accuracy. Parameter-averaging methods should be compared with the compressed models (pruned or distilled). In the 20-client CIFAR-100 experiment, the ensemble corresponds to roughly 20× the capacity of a single 3-layer CNN, and each client specializes on 5 classes. In the homogeneous setting this is equivalent to a 20-model ensemble trained on the same distribution. Results were verified across multiple seeds, and the provided run_code.bat script ensures reproducibility.

---

> > ### Author Response · Authors · 2025-11-17
> > **Weakness 4 Response**
> >
> > **Comments on Novelty (W4):**
> > We appreciate the reviewer’s suggestions on how to better highlight our contributions but we want to emphasize one point that other reviewers also seemed to miss.
> > Our goal is not merely to introduce a new method that outperforms FedOV, but to make a broader argument in favor of this emerging ensemble-based paradigm as a viable alternative to the dominant parameter-averaging approaches in federated learning. FedOV introduced the basic paradigm but did not analyze it in depth. Our contribution is to extend this perspective by improving its main weaknesses, namely scalability and OSR performance, and by showing its untapped potential for addressing communication and heterogeneity challenges that parameter averaging has not resolved. We believe future work can push this direction much further as more researchers become aware of its promise.

---

### Note · Authors · 2025-11-19

I have read and agree with the venue's withdrawal policy on behalf of myself and my co-authors.